

# Cost-efficient enactment of stream processing topologies

Christoph Hochreiner[1], Michael Vögler[2], Stefan Schulte[1] and Schahram Dustdar[1]

[1] Distributed Systems Group, TU Wien, Vienna, Austria
[2] TU Wien, Vienna, Austria

## ABSTRACT

The continuous increase of unbound streaming data poses several challenges to established data stream processing engines. One of the most important challenges is the cost-efficient enactment of stream processing topologies under changing data volume. These data volume pose different loads to stream processing systems whose resource provisioning needs to be continuously updated at runtime. First approaches already allow for resource provisioning on the level of virtual machines (VMs), but this only allows for coarse resource provisioning strategies. Based on current advances and benefits for containerized software systems, we have designed a cost-efficient resource provisioning approach and integrated it into the runtime of the Vienna ecosystem for elastic stream processing. Our resource provisioning approach aims to maximize the resource usage for VMs obtained from cloud providers. This strategy only releases processing capabilities at the end of the VMs minimal leasing duration instead of releasing them eagerly as soon as possible as it is the case for threshold-based approaches. This strategy allows us to improve the service level agreement compliance by up to 25% and a reduction for the operational cost of up to 36%.

## INTRODUCTION

Due to the transition toward a data-centric society, today's stream processing engines (SPEs) need to deal with a continuous increase of unbound streaming data regarding volume, variety, and velocity (*McAfee et al., 2012*). Currently, this growth in data is mainly driven by the rise of the internet of things (IoT) (http://www.gartner.com/newsroom/id/3165317). Sensors, which represent a vital part of the IoT, emit a huge volume of streaming data that needs to be processed to provide additional value to users or to trigger actions for IoT devices or other services, e.g., to handle user notifications. Furthermore, many scenarios call for data processing in near real-time, which requires the application of SPEs like System S (*Gedik et al., 2008*), Apache Storm (*Toshniwal et al., 2014*), Heron (*Kulkarni et al., 2015*), or Apache Spark (*Zaharia et al., 2010*). State-of-the-art SPEs provide the user with an extensive set of APIs to design and enact stream processing topologies. These topologies represent a composition of different stream processing operators, like filters, transformations, or other operations, which are required to process data (*Gedik et al., 2008*).

Although SPEs are highly efficient regarding data processing, they struggle with varying volumes of data over time (*Hochreiner et al., 2015*). Because most SPEs operate on a fixed

Corresponding author
Christoph Hochreiner,
c.hochreiner@infosys.tuwien.ac.at

amount of computational resources, e.g., on clusters, they cannot adapt to changes of the data volume at runtime (*Hochreiner et al., 2016a*). One solution for this issue is the over-provisioning of computational resources so that the SPE can process any amount of incoming data while complying with given service level agreements (SLAs). While this approach ensures a high level of SLA compliance, it is not cost-efficient because the provisioned computational resources are not used most of the time. The more economically feasible approach to this challenge is under-provisioning, where an SPE is equipped with computational resources to cover most of the incoming data scenarios. However, in the case of under-provisioning, the SPE may cover most scenarios, but it may also violate SLAs in some high load scenarios, due to a delay in the data processing.

Based on the Cloud Computing paradigm (*Armbrust et al., 2010*), a more promising provisioning approach, namely elastic provisioning for stream processing systems, emerged in recent years (*Satzger et al., 2011*; *Gedik et al., 2014*; *Heinze et al., 2015*; *Lohrmann, Janacik & Kao, 2015*; *Xu, Peng & Gupta, 2016*). This approach allows the SPE to lease computational resources on-demand whenever they are required. Resources can be released again as soon as they are not needed anymore. This approach allows for the cost-efficient enactment of stream processing topologies while maintaining high SLA compliance (*Hochreiner et al., 2016a*). Up to now, most elastic provisioning approaches only consider virtual machines (VMs) as the smallest entity for leasing and releasing of computational resources. This approach is perfect applicable for private clouds, where the only objective of resource provisioning algorithms is resource-efficiency, and there is no need to consider any billing aspects or billing time units (BTUs). A BTU defines the minimum leasing duration for computational resources, e.g., VMs, and often amounts to 1 h like on Amazon EC2 (https://aws.amazon.com/ec2/pricing/). The concept of the BTU means that the user has to pay for each started hour, regardless of how many minutes the VM is used. Because of the BTU, the repeated leasing and releasing of VMs may result in even higher cost than an over-provisioning scenario (*Genaud & Gossa, 2011*), because releasing a VM before the end of the BTU results in a waste of resources.

To address this shortcoming, this paper considers an additional resource abstraction layer on top of the VMs, to allow for more fine-grained elastic provisioning strategies with the goal to ensure the most cost-efficient usage of the leased resources while respecting given SLAs. This additional layer is realized by applying the recent trend toward containerized software components, i.e., containerized stream processing operators. The containerization provides several advantages regarding deployment and management of computational resources. Besides the smaller granularity compared to VMs, containerized stream processing operators also allow for a faster adaption of the stream processing topology on already running computational resources. An additional layer of containers also enables reusing already paid computational resources, i.e., resources can be utilized for the full BTU. Today, frameworks like Apache Mesos (*Hindman et al., 2011*), Apache YARN (*Vavilapalli et al., 2013*), Kubenetes (https://kubernetes.io) or Docker Swarm (https://docs.docker.com/swarm/) provide the functionality to deploy containerized applications on computational resources. These frameworks rely on simple principles like random deployment, bin-packing, or equal distribution to deploy

containers across multiple hosts. Although these approaches work well for most use cases, the resource usage efficiency for the underlying VMs in terms of their BTUs can be improved as we are going to show in our work.

In this paper, we propose an elastic resource provisioning approach which ensures an SLA-compliant enactment of stream processing topologies while maximizing the resource usage of computational resources and thus minimizing the operational cost, i.e., cost for computational resources and penalty cost for delayed processing, for the topology enactment. The results of our evaluation show that our approach achieves a cost reduction of about 12% compared to already existing approaches while maintaining the same level of quality of service. The main contributions of our work are the following:

- We propose a BTU-based resource provisioning approach, which only performs scaling activities when they are required to cope with the data volumes or when a downscaling operation can achieve a resource cost reduction while maintaining the same SLA compliance level.
- We extend the VISP Runtime (*Hochreiner et al., 2016b*) to support our BTU-based resource provisioning approach. The VISP Runtime is designed to serve as a test environment for resource provisioning mechanism and for this work we have not only implemented the BTU-based approach, but also refined the monitoring infrastructure to collect the information required for our approach.
- We implement a real-world scenario from the manufacturing domain and evaluate the BTU-based approach against a threshold-based baseline.

The remainder of this paper is structured as follows: first, we provide a motivational scenario, discuss the system architecture and present the derived requirements in "Motivation." Based on these requirements we then provide the problem definition for the optimization problem in "Problem Definition," which leads to our optimization approach presented in "Optimization Approach." In "Evaluation," we describe our evaluation setup and in "Results and Discussion," we present the evaluation results and their discussion. "Related Work" provides an overview on the related work, before we conclude the paper in "Conclusion."

## MOTIVATION

### Motivational scenario

In the following paragraphs, we describe a data stream processing scenario from our EU H2020 project Cloud-based Rapid Elastic Manufacturing (CREMA) (*Schulte et al., 2014*). Figure 1 shows a stream processing topology, which is composed of nine different stream processing operator types (O1–O9) that process the data originating from three different sources (S1, S2, S3). Each of the operator types performs a dedicated operation to transform the raw data from manufacturing machines into value-added and human-readable information. The information from the data sources is used to monitor three different aspects, like the availability of the manufacturing machines or the machine temperature to avoid overheating of the machines and assess their overall equipment

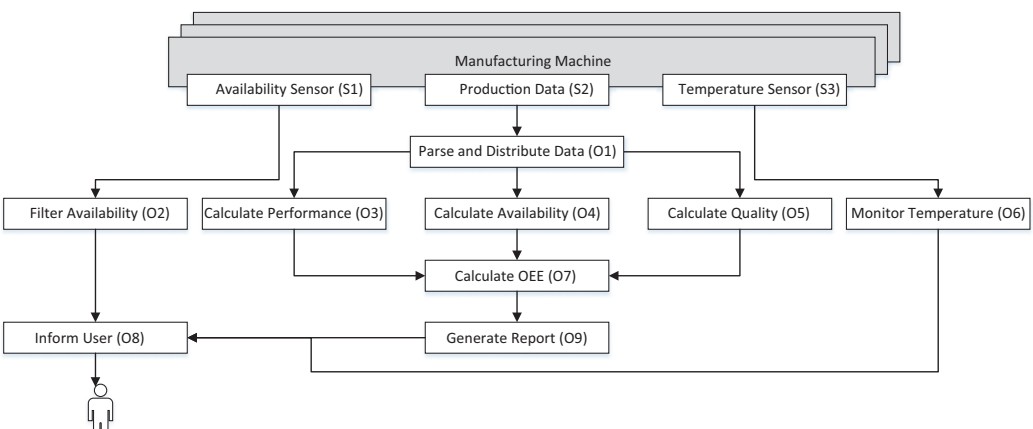

**Figure 1 Stream processing topology from the manufacturing domain.**

effectiveness (OEE). In this scenario, we have two different types of data sources. The first type of data source are sensors, i.e., S1 and S3, which emit machine-readable data and can be directly accessed via an API. The second type of data, e.g., S2, is a video feed, which scans a display of the manufacturing machines because some information is not directly accessible via an API. This information needs additional preprocessing to transform the data into machine-readable data.

The *Availability Sensor* (S1) emits the current status, i.e., available, defect or planned downtime, of the manufacturing machine every 2 s. This information is then filtered by the *Filter Availability* (O2) operator, which generates warnings for each new downtime incident of a specific manufacturing machine. The warning is then forwarded to the *Inform User* (O8) operator, which informs a human supervisor of the machines.

The second data source is the *Production Data* (S2), which is obtained by a video stream, i.e., an image taken every 10 s. This image contains different production-related information, such as the amount of produced goods and needs further processing, e.g., by optical character recognition (OCR), to extract machine-readable information. The *Parse and Distribute Data* (O1) operator distributes the information to the three operators O3, O4, O5 that calculate the different components of the OEE value. These individual components are then united by the *Calculate OEE* (O7) operator and afterwards forwarded to the *Generate Report* (O9) operator, which generates a PDF-report every minute. This report aggregates the information of all monitored machines and is forwarded once every minute to the *Inform User* (O8) operator.

The *Temperature Sensor* (S3) emits the temperature twice every second. This information is processed by the *Monitor Temperature* (O6) operator, which triggers a warning whenever the temperature exceeds a predefined threshold. This warning is then also forwarded to the *Inform User* (O8) operator to inform the human supervisor.

Due to the different levels of complexity of the operations, each of these operator types has different computational resource requirements, e.g., CPU or memory. Some of the operators, e.g., the Parse and Distribute Data operator type, require more resources

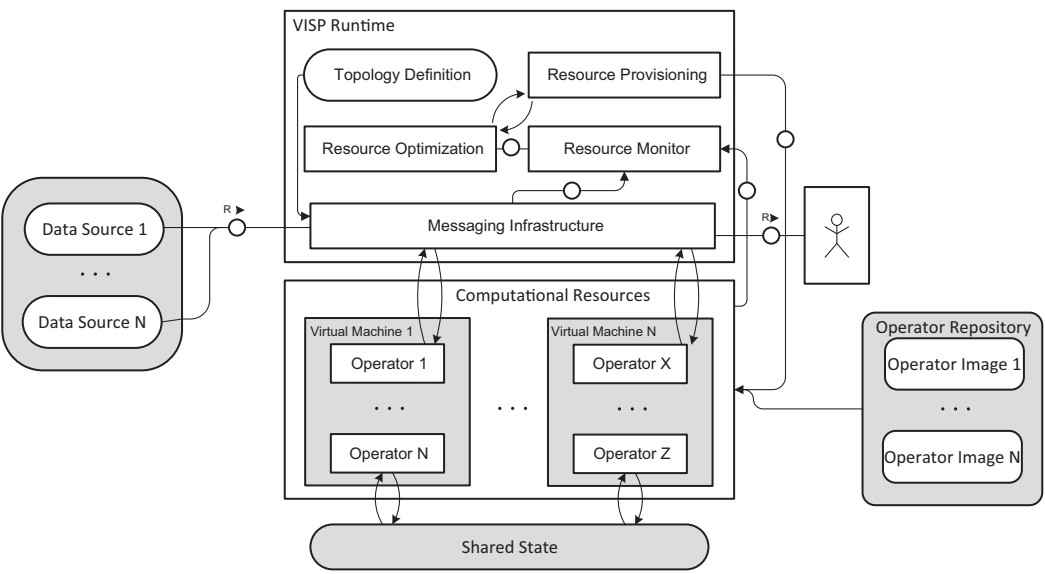

**Figure 2 VISP stream architecture.**

for processing one data item than others, like the Filter Availability. Besides the computational requirements, each operator type is also assigned with specific service level objectives (SLOs), like the maximal processing duration of one single data item. These SLOs are monitored, and whenever one operator type threatens to violate the imposed SLA, the system needs to provide more computational resources for data processing.

## System architecture

To enact the stream processing topology from the motivational scenario, it is required to instantiate it on an SPE. For our work at hand, we are extending the VISP ecosystem, which was introduced in our previous work (*Hochreiner et al., 2016b*). VISP represents an SPE, which is capable of provisioning computational resources on demand to adapt to the incoming load from data sources. VISP is composed of different components to cover the whole lifecycle of the stream processing topology enactment. Figure 2 shows a subset of these components, which are relevant for enacting the topology. For a detailed description of the components, please refer to our previous work (*Hochreiner et al., 2016b*) or to the online documentation of the VISP Ecosystem (https://visp-streaming. github.io), which is available under the Apache 2.0 License.

The primary task of the SPE, i.e., VISP Runtime, is to process data originating from data sources (on the left side of the figure) to obtain value-added data for users (on the right side of the figure) based on the Topology Definition. The actual data processing is conducted by Operators, which are deployed on computational resources, e.g., VMs, provided by an infrastructure as a service environment. Each operator type is instantiated from dedicated operator images hosted on an external operator repository. To instantiate a specific operator instance on any host for the first time, the operator image needs to be downloaded from the registry, which takes a certain amount of time, depending on the size of the operator image. After the first instantiation of the operator

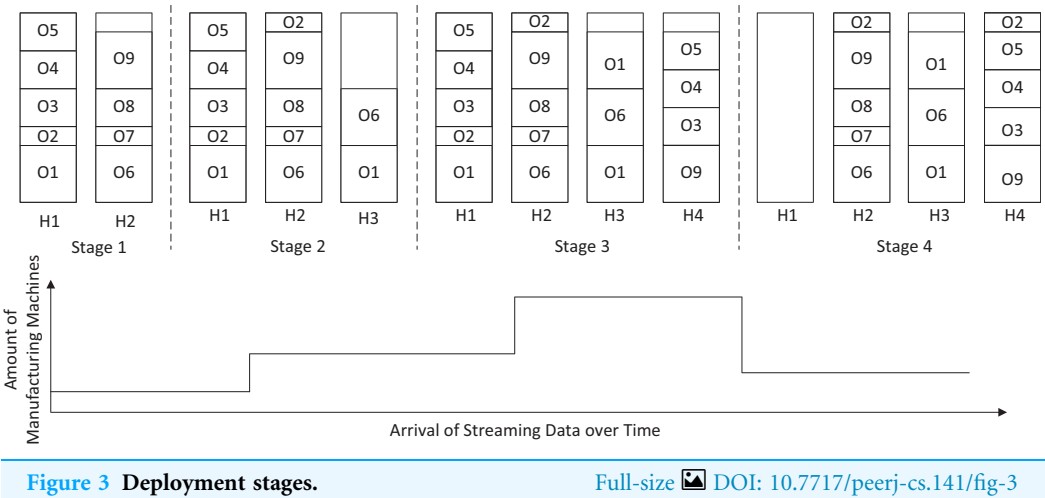

**Figure 3** Deployment stages.    

type, the operator image is cached locally on the host to speed up the instantiation of future instances. Each operator type is also assigned with individual SLAs whereas each SLA consists of different SLOs. The first SLO is the maximum processing duration for one data item and ensures the near real-time processing capabilities of the stream processing topology. The second SLO describes the minimal resource requirements that are needed to instantiate the stream processing operator. These requirements are represented by the minimum amount of memory, i.e., Memory in MegaByte (MB), and the number of CPU shares.

For the enactment of a stream processing topology, each Operator from the topology is represented by at least one, but up to arbitrarily many Operators. These Operators fetch the data from the Messaging Infrastructure according to the Topology Definition, process it and push it back for further processing steps. The remaining components of the VISP Runtime are in charge of monitoring the load on the Messaging Infrastructure as well as on the Operators. This monitoring information is then used by the Resource Optimization component to evaluate whether operator types need to be replicated to deal with the incoming load. Finally, the Resource Provisioning component is in charge of deploying and un-deploying Operators on computational resources.

## Enactment scenario

During the enactment, the stream processing operators need to deal with streaming data from a varying amount of manufacturing machines, as shown in Fig. 3 at the bottom. This varying data volume requires the SPE to adapt its processing capabilities, i.e., the number of operator instances for specific operator types, which are hosted on an arbitrary amount of hosts, e.g., H1–H4 in Fig. 3, on demand to comply with the SLAs. Nevertheless, the SPE aims at minimizing the needed number of hosts, since each host amounts for additional cost, by using an optimal deployment.

The enactment of our motivational scenario is partitioned into different stages, with a varying number of running manufacturing machines in each stage. At the beginning of stage 1, each operator is deployed once across the two hosts H1–H2. Since the volume

of streaming data increases after some time, the SPE needs to adapt the processing capabilities by deploying replicas of the operator types O1, O2 and O6 in stage 2. These operator instances are hosted on a new host H3 because the two already existing hosts cannot cope with the additional operator instances. Because the amount of data increases again in stage 3, the SPE needs to replicate further operators to comply with the SLAs. Although the second replication of the operator type O1 is feasible on the currently available resources, the SPE is required to lease a new host for the additional operator instances of types O3, O4, O5, and O9.

At the end of stage 3, H1–H2 meet the end of their BTU. Therefore, the SPE evaluates whether some of the replicated operators can be removed again without violating the SLAs. Because the amount of data is decreasing after stage 3, the system can remove (O1, O3, O4, and O5) or migrate (O2) some of the operator instances to other hosts. This leads to the situation that no operator instances are running on host H1 at the end of its BTU and the SPE can accordingly release H1, while host H2 needs to be leased for another BTU.

## Requirements

Based on our motivational scenario, we have identified several requirements which need to be addressed by the optimization approach.

### SLA compliance

The first requirement is SLA compliance in terms of maximum processing duration, for data that is processed by the stream processing topology. This compliance is the overall goal that needs to be met, regardless of the actual incoming data rate.

### Cost efficiency

The second requirement is the cost efficiency for the enactment. This requirement asks for a high system usage of leased resources and an efficient usage of cloud resources, especially regarding their BTU.

### Optimization efficiency

The optimization efficiency requirement can be split into two different aspects. The first aspect is the solution of the optimization problem presented in "Problem Definition." Because this optimization problem is NP-hard (see Optimization Problem), it is required to devise heuristics to achieve a time- and resource-efficient optimization approach. The second aspect is that the optimization needs to minimize the number of reconfigurations, e.g., scaling operations, for the stream processing topology because each reconfiguration activity has a negative performance impact on the data processing capabilities.

## PROBLEM DEFINITION

### System model and notation

The system model is used to describe the system state of the individual operator types that form the stream processing topology as well as the used computational resources. The

individual operator types are represented by $O = \{1, \ldots, o^{\#}\}$, where $o \in O$ represents a specific operator type. Each operator type $o$ is assigned with minimal resource requirements $o_{\text{cpu}}$ and $o_{\text{memory}}$ which need to be met to instantiate an operator on any host. At runtime, each operator type is represented by at least one, but up to arbitrary many operator instances, which are described by the set $I = \{1, \ldots, i^{\#}\}$, whereas each $i_{\text{type}}$ is assigned to a particular operator type $o \in O$.

This set of operator instances $I$ is running on arbitrarily many hosts that are represented by the set $H = \{1, \ldots, h^{\#}\}$, whereas each host hosts a subset of $I$. Each of these hosts is furthermore assigned with a set of attributes. The attributes $h_{\text{cpu}}$ and $h_{\text{memory}}$ represent the overall computational resources of the host, and the attributes $h_{\text{cpu}^*}$ and $h_{\text{memory}^*}$ represent the remaining computational resources at runtime. The attributes $h_{\text{cpu}^*}$ and $h_{\text{memory}^*}$ are decreased for every operator instance $i$ on the specific host $h$ and can be used to determine if it is possible to deploy an additional operator instance on this particular host $h$. The attribute $h_{\text{cost}}$ represents the cost for the host, which needs to be paid for each BTU. The attribute $h_{\text{BTU}^*}$ represents the remaining, already paid, BTU time. To represent the different startup times between cached and non-cached operator images, each host furthermore denotes a set of images $h_{\text{img}}$. This set contains all operator images $o \in O$, which are cached on this particular host. Each operator type is assigned a specific image, whose identifier is identical to the name of the operator type.

Besides the fundamental operator type attributes for instantiating operators, there is also a set of attributes, which is used to ensure the SLA compliance for data processing. Each operator type is assigned with an estimated data processing duration $o_{\text{slo}}$, that represents the time to process one data item and pass it on to the following operator type according to the stream processing topology. The $o_{\text{slo}}$ value is recorded in an optimal processing scenario, where no data item needs to be queued for data processing. Since the SLO $o_{\text{slo}}$ only presents the expected processing duration, we also denote the actual processing duration for each operator $o_{\text{d}}$ and the amount of data items $o_{\text{queue}}$ that are queued for a particular operator type for processing.

In addition to the current $o_{\text{d}}$, the system model also considers previous processing durations. Here we consider for each operator type $o$, the last $N$ processing durations $o_{\text{d}}$ denoted as $o_{\text{d1}}$ to $o_{\text{dN}}$, whereas each of the values gets updated after a new recording of the $o_{\text{d}}$, i.e., $o_{\text{d1}}$ obtains the value of $o_{\text{d}}$ and $o_{\text{d2}}$ obtains the value of $o\text{d}_1$, etc. If the actual processing duration $o_{\text{d}}$ takes longer than the SLO $o_{\text{slo}}$, penalty cost $P$ accrue to compensate for the violated SLAs each time a violation $v \in V$ occurs.

Furthermore, we denote two operational attributes for each operator type. The attribute $o_{\#}$ represents all current instances, i.e., the sum of all instances of the operator type $o$, and the attribute $o_{\text{s}}$ represents all already executed scaling operations, both upscaling and downscaling, for a specific operator type. Last, we also denote the current incoming amount of data items as $DR$.

## Optimization problem

Based on the identified requirements in "Requirements," we can formulate an optimization problem as shown in Eq. (1). The goal of this optimization problem is

to minimize the cost for the topology enactment while maintaining given SLOs. This equation is composed of four different terms, which are designed to cover the different requirements. The first term represents the cost for all currently leased hosts by multiplying the number of all currently leased hosts with the cost for a single host. The second and third term are designed to maximize the resource usage on all currently leased hosts regarding the CPU and memory. The last term ensures the SLA compliance of the deployment, due to the penalty cost, which accrue for each SLO violation.

Although the solution of this optimization problem provides an optimal solution for a cost-efficient deployment, it is not feasible to rely on the solution of this problem due to its complexity. To define the complex nature of this problem, we are going to provide a reduction to an unbounded knapsack problem (*Andonov, Poirriez & Rajopadhye, 2000*), which is known to be NP-hard.

$$
\begin{aligned}
\text{Min} \quad & h^{\#} \cdot h_{\text{cost}} \\
& + \frac{\sum_{h \in H} h_{\text{cpu}} - \sum_{i \in I \cap i_{\text{type}} = o} o_{\text{cpu}}}{\sum_{h \in H} h_{\text{cpu}}} \\
& + \frac{\sum_{h \in H} h_{\text{memory}} - \sum_{i \in I \cap i_{\text{type}} = o} h_{\text{memory}}}{\sum_{h \in H} h_{\text{memory}}} \\
& + \sum_{v \in V} v \cdot P
\end{aligned}
\tag{1}
$$

### Definition of knapsack problem

The unbounded knapsack problem assumes a knapsack, whose weight capacity is bounded by a maximum capacity of $C$ and a set of artifacts $A$. Each of these artifacts $a$ is assigned with a specific weight $a_{\text{w}} > 0$ as well as a specific value $a_{\text{v}} > 0$ and can be placed an arbitrary amount of times in the knapsack. The goal is to find a set $A1$ of items, where $\sum_{a \in A} a_{\text{w}} \leq C$ and $\sum_{a \in A} a_{\text{v}}$ is maximized.

### NP-hardness of the optimization problem

For our reduction, we assume a specific instance of our optimization problem. For this specific instance, we assume that the number of hosts is fixed and that each of the operators has the same memory requirements $o_{\text{memory}}$. Furthermore, we define the value of a specific operator by the amount of data items $o_{\text{queue}}$ that are queued for a specific operator type, i.e., the more items need to be processed, the higher is the value for instantiating a specific operator.

Based on this specific instance of the optimization problem, we can build an instance of the unbounded knapsack problem, where the maximum capacity $C$ is defined by the maximum amount of CPU resources on all available hosts $\sum_{h \in H} h_{\text{cpu}}$, the weight $a_{\text{w}}$ of the artifacts $a$ is defined by the CPU requirements $o_{\text{cpu}}$ of one operator and the value $a_{\text{v}}$ of the artifact is defined by the number of items waiting on the operator type-specific queue $o_{\text{queue}}$.

Because a specific instance of our optimization problem can be formulated as a knapsack problem, we can conclude that our optimization problem is also NP-hard. This concludes that there is no known solution which can obtain an optimal solution in polynomial time. Since this conclusion conflicts with the third requirement given in "Requirements," we decided to realize a heuristic-based optimization approach, which can be solved in polynomial time.

## OPTIMIZATION APPROACH

The overall goal our optimization approach is to minimize the cost for computational resources and maximize the usage of already leased VMs while maintaining the required QoS levels. Therefore, we apply an on-demand approach to reduce the deployment and configuration overhead, i.e., instantiating and removing additional operator instances, and minimize the computational resources required for finding an optimal deployment configuration. Due to our emphasis on the BTUs of VMs, we call our approach BTU-based approach in the remainder of this paper.

### Ensure sufficient processing capabilities

To avoid penalty cost, our approach continuously evaluates the SLA compliance of the stream processing topology. Whenever the individual processing duration $o_d$ of a particular operator type $o$ exceeds or threatens to exceed the maximum allowed processing duration $o_{slo}$ according to the *Upscaling Algorithm* as shown in Algorithm 1, the upscaling procedure for the specific operator type is triggered.

This upscaling procedure consists of several steps, as depicted in Fig. 4. The first task is to evaluate if any of the currently running hosts offers enough computational resources to host the additional instance of the specific operator. Therefore, we apply the *Host Selection Algorithm*, as described in Algorithm 2, for every currently running host to obtain a utility value for the host. Assuming that there is at least one host with a positive utility value, the host with the best utility value is selected to deploy the new operator instance, and the upscaling procedure is finished.

When no host with a positive utility value is available, i.e., no hosts offers enough computational resources to instantiate a new instance for the required operator type, there are two possibilities to obtain the required computational resources. The first possibility is to scale down existing operators when they are not required anymore. We therefore apply the *Operator Selection Algorithm*, as described in Algorithm 3 and discussed in "Algorithms." If there is any operator type that can be scaled down, an operator instance of this operator type will be scaled down to free resources for the upscaling operation. When there are no operator types which can be scaled down, i.e., all operators are needed for SLA-compliant data stream processing, the second possibility is applied where the SPE leases a new host.

As soon as the resources are either provided by scaling down another operator type or the new host is running, the SPE deploys the required operator instance and finishes the upscaling procedure.

**Algorithm 1  Upscaling Algorithm**

1:   **function** UPTRIGGER(o,N)

2:      **if** $o_d > o_{\text{slo}}$ **then**

3:        upscaling = 1

4:      **end if**

5:      observationMean $= \frac{1}{N} * \sum_{i=1}^{N} i$

6:      durationMean $= \frac{1}{N} * \sum_{i=1}^{N} o_{d_i}$

7:      $\beta = \dfrac{\sum_{i=1}^{N}(i - \text{observationMean}) * (o_{d_i} * \text{durationMean})}{\sum_{i=1}^{N}(i - \text{observationMean})^2}$

8:      $\alpha$ = durationMean $- \beta$ * observationMean

9:      predictedDuration $= \alpha + \beta * (N + 1)$

10:     **if** predictedDuration $> o_{\text{slo}}$ **then**

11:       upscaling = 1

12:     **end if**

13:     **if** upscaling = 0 **then**

14:       **return** 0

15:     **end if**

16:     **if** $o_{\text{queue}} >$ scalingThreshold **then**

17:       **return** 1

18:     **end if**

19:     **return** 0

20:  **end function**

## Optimize resource usage

To minimize the cost of computational resources, the optimization approach aims at using the leased resources as efficient as possible. This means that the SPE uses all paid resources until the end of their BTUs and evaluates shortly before, i.e., within the last 5% of the BTU, whether a host needs to be leased for another BTU, i.e., the resources are still required, or if the host can be released again.

To release hosts, as shown in Fig. 5, all operator instances running on the designated host which is targeted to be shut down, need to be either released or migrated to other hosts. This releasing procedure consists of three phases. The first phase is a simulation phase, where the optimization approach creates a downscaling plan to evaluate whether the downscaling and migration is actually feasible. Hereby, the optimization approach applies the *Operator Selection Algorithm* for all operator types, which have running instances on this host and obtain their utility value. If any of the operator types has a positive utility value, all operator instances (up to 20% of all operator instances for the specific type) running on this host are marked to be released. The 20%-threshold for the operator instances is in place to avoid any major reconfigurations for a single operator type, since it may be the case that all operator instances for the operator type are running on this host and after the downscaling there would be not sufficient operator instances left

 

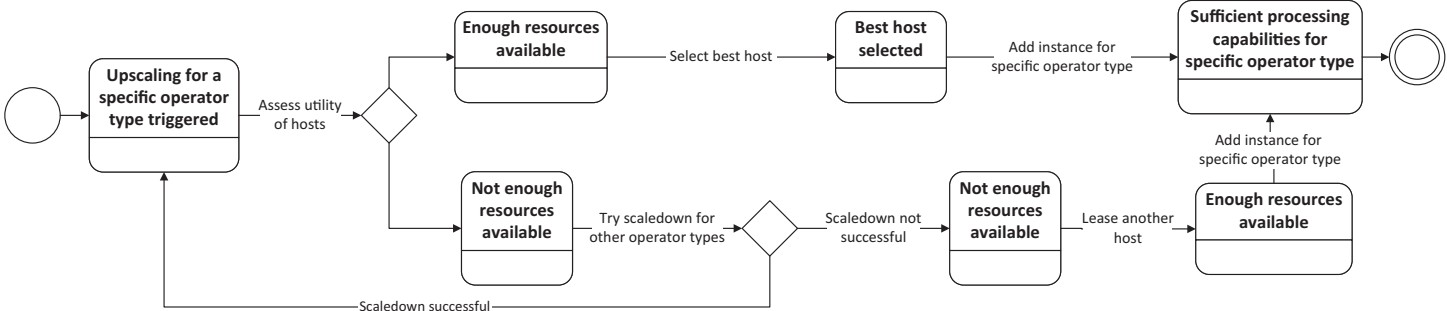

**Figure 4 Upscaling procedure for a specific operator type.**

---

**Algorithm 2  Host Selection Algorithm**

1: **function** UP(h,o)
2:     feasibilityThreshold = min$((h_{cpu}{}^*/o_{cpu}), (h_{memory}{}^*/o_{memory}))$
3:     **if** feasibilityThreshold < 1 **then**
4:         **return** −1
5:     **end if**
6:     remainingCPU = $h_{cpu}{}^* − o_{cpu}$
7:     remainingMemory = $h_{memory}{}^* − o_{memory}$
8:     $\text{difference} = \left| \dfrac{\text{remainingCPU}}{h_{cpu}} - \dfrac{\text{remainingMemory}}{h_{memory}} \right|$
9:     $\text{suitability} = \dfrac{\text{difference}}{\text{feasibilityThreshold}}$
10:     **if** $s \in h_{img}$ **then**
11:         suitability = suitability * CF
12:     **end if**
13:     **return** suitability
14: **end function**

---

which would trigger again the upscaling procedure. For those operator instances which cannot be shut down, the procedure simulates whether they can be migrated to other hosts. This simulation uses the upscaling procedure for operator types, as described in "Ensure Sufficient Processing Capabilities." The only difference is that the host which is targeted to be shut down, is omitted as a suitable host.

If the simulation renders no feasible downscaling plan, the host is leased for another BTU and the downscaling procedure is finished. In case there is a downscaling plan, the operators are released in phase two and if any migration is required, the upscaling procedure for operator types is triggered based on the simulation in phase three. When all operator instances are successfully removed (scaled down or migrated), the shutdown of the host is initialized. In the unlikely event that the downscaling plan could not be executed, i.e., the operator instance migrations fail, the host also needs to be leased for another BTU.

---

**Algorithm 3** Operator Selection Algorithm

1: **function** DOWN(o)
2:    **if** $o_\#$ < 2 **then**
3:      **return** −1
4:    **end if**
5:    $instances = \dfrac{o_\# - \min(o_\# \in O)}{\max(o_\# \in O) - \min(o_\# \in O)}$
6:    $delay = \frac{o_d}{o_{slo}} * (1 + P)$
7:    $scalings = \dfrac{o_s}{\sum_{o_s \in O} o_s}$
8:    **if** $o_{queue}$ < 1 **then**
9:      queueLoad = QL
10:    **else**
11:      queueLoad = 0
12:    **end if**
13:    **return** 1 + W1 * instances + W2 * queueLoad − W3 * delay − W4 * scalings
14: **end function**

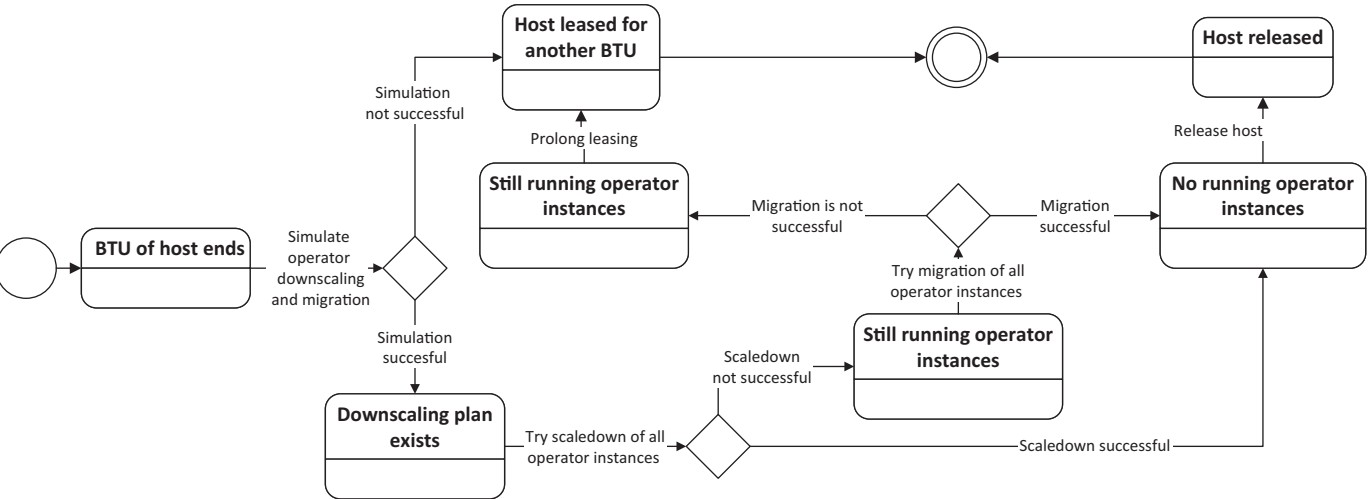

**Figure 5 Downscaling procedure for a host.**

## Algorithms

To realize our BTU-based provisioning approach, we have devised three algorithms, which are discussed in detail in this section. These three algorithms realize individual tasks for the upscaling and downscaling procedures as shown in Figs. 4 and 5. Algorithm 1 ensures the SLA compliance of the individual operator types on a regular basis by interpreting the monitoring information of the VISP Runtime. The other two algorithms are only triggered if a new operator instance needs to be started or when there is a shortage of free computational resources. These two algorithms analyze the SLA compliance and resource

usage on demand at specific points in time and identify the most suitable host for upscaling (Algorithm 2) or potential operator types, which can be scaled down (Algorithm 3). Although these algorithms do not represent the core functionality of the resource provisioning approach, they are still essential to identify required upscaling operations and choose the optimal degree of parallelism per operator whereas the overall cost-reduction and reconfiguration is represented by the downscaling procedure shown in Fig. 5. The remainder of this section discusses the structure and rationale of these three algorithms in detail.

The *Upscaling Algorithm* as listed in Algorithm 1 is used to evaluate whether any operator needs to be scaled up. This algorithm is executed on a regular basis for each operator type $o$ and either returns 0, if the current stream processing capabilities are enough to comply with the SLAs, or 1 if the operator type needs to be scaled up. Therefore, this algorithm considers, on the one hand, the current processing duration of the operator (Line 2) and, on the other hand, the trend of the previous processing durations. For the trend prediction, we apply a simple linear regression for the last $N$ observations, based on the linear least squares estimator (Lines 5–9). If the current duration $o_d$ or the predicted duration is higher than the SLO $o_{slo}$, we consider the operator type to be scaled up (Line 10). Before we trigger the upscaling operation, we apply an additional check if the upscaling operation is required.

The stream processing topology may retrieve short-term data volume peaks, e.g., due to short network disruptions. These peaks would not require any additional computational resources, because they would be dealt with after a short time with the already available processing capabilities. Nevertheless, the upscaling algorithm would trigger the upscaling procedure, because it would detect the processing delay. Therefore, the algorithm also considers the current load of data items $o_{queue}$ before scaling up by checking whether the amount of queued items for processing exceeds a *scalingThreshold* (Lines 13–16).

Algorithm 2, i.e., the *Host Selection Algorithm*, is used to rank all currently leased hosts according to their suitability to host a new operator instance of a particular operator type. Therefore, the algorithm evaluates for each host $h$ whether a new instance of the required operator type $o$ could be hosted on that specific host at all. Here, the algorithm considers both, the CPU and memory requirements, and derives the maximum amount of instances that can be hosted. If this value is less than 1, i.e., there are no resources left for a single additional operator instance, the function returns a negative value. The first check evaluates the feasibility of deploying a new operator instance on the host (Lines 2–5). In a second stage, this algorithm evaluates the suitability of this host. Here the algorithm simulates the resource usage of the host, assuming the operator instance would be deployed on the host. The overall goal is an equal distribution of CPU and memory usage across all hosts, to avoid situations where hosts maximize their CPU usage, but hardly use any memory and vice versa. Therefore, the algorithm calculates the difference between the normalized CPU usage and memory usage, whereas a lower value represents a better ratio between CPU and memory and therefore a better fit (Lines 6–9). Besides the equal distribution of memory and CPU on the individual hosts, we also want to distribute

the operators equally among all currently leased hosts. The assigned CPU $o_{cpu}$ and memory $o_{memory}$ attributes only represent the resources which are guaranteed for the operators. This allows operators to use currently unused resources of the hosts based on a first come first service principle. To maximize the usage, we aim for an equal distribution of the unassigned resources, i.e., $h_{cpu^*}$ and $h_{memory^*}$, which can be used by the operators to cover short-term data volume peaks without any reconfigurations required. This aspect is covered by dividing the *difference* value by the *feasibility* value to prefer those hosts which are least used (Line 9). Last, we also consider the deployment time aspect for a particular operator type. Here, we prefer those hosts, which have already the operator image cached. While such operator images may be rather small for SPEs which operate on a process or thread level, like Apache Storm, these images can reach up to 100 MB for containerized operators. This requires some time to download the operator images from an external repository. In order to distinguish hosts, which have a cached copy of the operator image from those hosts that do not have a cached copy of the operator image, we multiply the *suitability* with a constant factor *CF* to create two different groups of hosts for the overall selection (Lines 10–12). For this constant factor, we recommend to use the value 0.01 which was also used in the remainder of our work. The value 0.01 was chosen to clearly distinguish these two groups, since the actual suitability values are always in the range of 0–1 based on the structure of the algorithm. Each of these group maintains their resource-based ordering, but we prioritize those hosts that provide a faster startup time due to the cached image, i.e., the group with lower values. The result of this algorithm is either a negative value for a host, i.e., the host can run the new operator instance, or a positive value, whereas the lowest value among several hosts shows the best suitability.

Algorithm 3, i.e., the *Operator Selection Algorithm*, is used to select operator types which can be scaled down without violating the SLOs. Therefore, this algorithm considers several static as well as runtime aspects of the operator types. The goal of the algorithm is to obtain a value which describes the suitability of a particular operator type to be scaled down. Whenever the value is negative, the operator type must not be scaled down, i.e., all operator instances for this type are required to fulfill the SLO.

First, the algorithm ensures that there is at least one operator instance for the given operator type (Lines 2–4). Second, the function considers the amount of all currently running instances for the specific operator type and normalizes it to obtain a value between 0 and 1 (Line 5). This normalization is carried out based on the maximal respectively minimal amount of instances for all operator types. This value represents the aspect that it is better to scale down an operator type with numerous operator instances because the scale down operation removes a smaller percentage of processing power compared to an operator type with fewer operator instances.

Furthermore, we consider the SLA compliance of the particular operator. Here, we consider the actual compliance for the processing duration and multiply it with the penalty cost as a weighting factor (Line 6). Since the penalty cost for the violation of a single data item is typically lower than 1, we add 1 to the penalty cost *P*. Whenever the processing duration $o_d$ takes longer than the SLO $o_{slo}$, the delay value will be less than 1,

but when there is any delay, the delay value can become arbitrarily high. The next value for consideration is the relative amount of scaling operations (both up and down) in contrast to the entire scaling operations (Line 7). Here, we penalize previous scaling operations because we want to avoid any oscillating effects, i.e., multiple up- and down-scaling operations for a specific operator. The last factor is the queueLoad. In the course of our evaluations, we have seen that the algorithm may take a long time to recover after a load peak, i.e., release obsolete operator instances as soon as the data is processed. This can be observed when the SPE is confronted with a massive data spike followed by a small data volume for some time. For this scenario, the heuristic discourages any downscaling operation due to the delay factor, which may be high due to the delayed processing of the data spike. To resolve this shortcoming, we introduce the queueLoad factor $QL$, which encourages the downscaling of an operator type, as soon as no data items are waiting in the incoming queue $o_{queue}$ (Lines 8–12). For $QL$ we recommend the use of the value 100 to clearly indicate that the operator type can be scaled down, regardless of the other values which are in the range of 0–1 for the *instances* and *scalings* value or significantly lower than 100 for the *delay* value. This value was selected based on a number of preliminary experiments prior to the actual evaluation where the data processing never took longer than 50 times of the intended processing duration.

Finally, we join the distinct aspects to obtain the overall utility value. While the number of instances and queueLoad represent a positive aspect to scale down an operator, all other aspects discourage a scaling operation. The instances and scalings value are normalized between 0 and 1 whereas the scalings value can exceed 1 if the data processing is delayed. Therefore, we introduce optional weights $W1$, $W2$, $W3$, and $W4$ for the different aspects, whereas the default value for each of these weights is 1 to treat all aspects with the same emphasis. The result is the utility value, which describes the suitability of the particular operator to be scaled down, whereas a higher value suggests a better suitability (Line 13).

# EVALUATION

## Evaluation setup

For our evaluation, we revisit our motivational scenario (see Motivation) and discuss the concrete implementation of this topology.

### Sensor types

First, we are going to discuss the sensors which emit the data items for our topology. In this topology, we consider three different sensor types, as listed in Table 1. Each of these sensor types generates a data item, with a particular structure, which can be only processed by a dedicated operator type, e.g., O1 for sensor type S2. Due to the different structure, the size of the data items also differs. The first and the last sensor type (S1 and S3) encode the information in plain text. This results in rather small data items with a size of 90–95 Bytes. The second sensor type encodes the information with an image and is therefore much larger, i.e., around 12,500 Bytes.

**Table 1** Sensor types.

|  | Emission rate/min | Size (Bytes) |
|---|---|---|
| Availability sensor (S1) | 5 | 95 |
| Production data (S2) | 1 | 12,500 |
| Temperature sensor (S3) | 10 | 90 |

**Table 2** Stream processing operator types.

|  | Processing duration (ms) | CPU shares | Memory (MB) | Storage (MB) | State | Outgoing ratio |
|---|---|---|---|---|---|---|
| Parse and Distribute Data (O1) | 1,500 | 660 | 452 | 89 | ✓ | 1:3 |
| Filter Availability (O2) | 600 | 131 | 524 | 68 | ✓ | 50:1 |
| Calculate Performance (O3) | 750 | 100 | 430 | 68 | ✓ | 1:1 |
| Calculate Availability (O4) | 750 | 83 | 502 | 68 | ✓ | 1:1 |
| Calculate Quality (O5) | 750 | 77 | 527 | 68 | ✓ | 1:1 |
| Monitor Temperature (O6) | 600 | 65 | 440 | 68 | ✓ | 100:1 |
| Calculate OEE (O7) | 700 | 46 | 464 | 68 | ✓ | 3:1 |
| Inform User (O8) | 500 | 74 | 466 | 68 | ✓ | 1:0 |
| Generate Report (O9) | 1,300 | 47 | 452 | 70 | ✓ | 300:1 |

## Operator types

The second important implementation aspect for the topology are the operators. Each of these operator types performs a specific task with specific resource requirements and specific processing durations. Table 2 lists all operator types which are used in this evaluation. Each operator is assigned a number of different performance as well as resource metrics. The resource metrics represent mean values across several topology enactments. The processing duration represents the average times which are required to process one specific data item as well as the time the data item is processed within the messaging infrastructure between the previous operator and the one in focus. The CPU metric represents the amounts of shares which are required by the operator when executed on a single core VM. The memory value represents the mean memory usage. This memory value accumulates the actual used memory by the operator instances and the currently used file cache, which results in a rather high value compared to the actual size of the operator image. The CPU metric and the memory metric are determined based on long-term recordings, whereas the stated value in the table is calculated by adding both the absolute maximum and the average value of all observations for a specific operator and dividing this value by 2. For the processing duration, we have conducted several preliminary evaluations, where the SPE is processing constant data volumes in a fixed over-provisioning scenario to avoid any waiting durations for the recordings.

For the storage operator, we have three different sizes. Because the majority of the processing operators only implement processing logic, the size of the images is the same. The only two exceptions are the Generate Report (O9) image, which also contains a PDF

generation library and the Parse and Distribute Data (O1) operator image, which also contains the tesseract binary, which is required to parse the images. Each of the stateful operators, as indicated in the table, can store and retrieve data from the shared state to synchronize the data among different data items and different instances of one operator type. The outgoing ratio describes whether a particular operator type consumes more data items than it emits, e.g., O7 combines three data items before it emits a combined one, or whether it emits more data items than it receives, e.g., O1 distributes the production information to three other operator types.

For our scenario, we have implemented nine different operators (https://github.com/visp-streaming/processingNodes) as Spring Boot (https://projects.spring.io/spring-boot/) applications, which are discussed in detail in the remainder of this section.

### Parse and distribute data (O1)

The Parse and Distribute Data operator type is designed to receive an image with encoded production data and parse this image to extract the information. For our implementation, we use the tesseract OCR engine (https://github.com/tesseract-ocr/tesseract) to parse the image and then the Spring Boot application forwards the machine-readable production data to the downstream operator types.

### Filter availability (O2)

Each manufacturing machine can have three different availability types: available, planned downtime, and defect. While the first two types represent intended behavior, the last type signals a defect and should be propagated to a human operator. This operator issues a warning for each new defect notification and filters all other data items.

### Calculate performance (O3)

The Calculate Performance operator type calculates the performance of the last reporting cycle, i.e., the time between two production data emissions. The actual performance is derived by the formula shown in Eq. (2) (*Nakajima, 1988*).

$$\text{performance} = \frac{\text{producedItems} \times \text{idealProductionTime}}{\text{reportingCycle}} \tag{2}$$

### Calculate availability (O4)

The Calculate Availability operator type represents the overall availability of the manufacturing machine from the beginning of the production cycle, e.g., the start of the evaluation. The availability is defined by the formula shown in Eq. (3) (*Nakajima, 1988*).

$$\text{availability} = \frac{\text{totalTime} - \text{scheduledDowntime} - \text{unscheduledDowntime}}{\text{totalTime}} \tag{3}$$

### Calculate quality (O5)

The Calculate Quality operator type represents the ratio between all produced goods against defect goods from the beginning of the production cycle. The quality is defined by the formula shown in Eq. (4) (*Nakajima, 1988*).

$$\text{quality} = \frac{\text{totalProducedGoods} - \text{totalDefectiveGoods}}{\text{totalProducedGoods}} \qquad (4)$$

*Monitor temperature (O6)*
The Monitor Temperature operator type filters all temperatures below a predefined threshold and issues a notification to the human operator for each new temperature violation.

*Calculate OEE (O7)*
The Calculate OEE operator synchronizes the upstream operations based on the timestamp of the initial data item and calculates the overall OEE value according to the formula in Eq. (5).

$$\text{OEE} = \text{availability} \cdot \text{performance} \cdot \text{quality} \qquad (5)$$

*Inform user (O8)*
The Inform User operator type forwards the notifications to a human user. In our evaluation scenario, this operator type only serves as a monitoring endpoint for the SLA compliance and all incoming data items are discarded at this operator type.

*Generate report (O9)*
The Generate Report operator aggregates multiple OEE values and generates a PDF report which aggregates a predefined amount of OEE values. This report is then forwarded to the user for further manual inspection.

## Evaluation deployment

For our evaluation, we make use of the VISP Testbed (*Hochreiner, 2017*), which is a toolkit of different evaluation utilities that support repeatable evaluation runs. The most notable component of this toolkit is the VISP Data Provider, which allows simulating an arbitrary amount of data sources. Furthermore, the Data Provider also allows defining different arrival patterns (see Data Arrival Pattern) to evaluate the adaptation possibilities of the VISP Runtime, in particular of its scaling mechanisms.

The evaluation runs are carried out in a private cloud running OpenStack (https://www.openstack.org), whereas the components are deployed on different VMs, as depicted in Fig. 6. The most relevant VM for our evaluation is the Infrastructure VM, which hosts the VISP Runtime as well as all other relevant services, like the Message Infrastructure, i.e., RabbitMQ (https://www.rabbitmq.com), the Shared State, i.e., Redis (http://redis.io) and the Data Storage, i.e., a MySQL (https://www.mysql.com) database.

For the topology enactment, the VISP Runtime leases (and releases) an arbitrary amount of VMs, i.e., Dockerhost VMs, on the private OpenStack-based cloud at runtime. These Dockerhost VMs are used to run the Operator Instances, which take care of the actual data processing as described in "System Architecture." The Operator Images, which are required to run the Operator Instances, are hosted on an external service, i.e.,

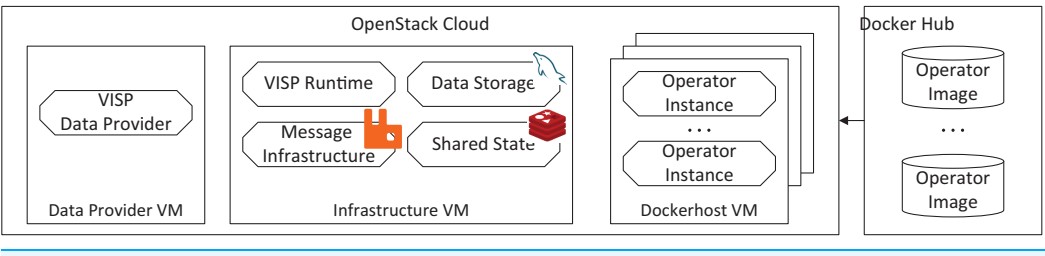

**Figure 6 Deployment for the evaluation scenario.**

Dockerhub (https://hub.docker.com). Finally, the Data Provider VM is in charge of simulating the data stream from the sensors, as described in "Sensor Types."

## Evaluation configuration

For the *scalingThreshold* used in Algorithm 1, we use the value 50. This value was selected to be high enough to allow for minimal hardware disturbances, e.g., moving data from memory to the hard drive, but low enough to react to small changes of the data volume. The concrete value was identified on a number of preliminary experiments, evaluating different thresholds in the range of 10–1,000 items, whereas the threshold 50 was identified as the most suitable value for our purpose. Regarding the individual weights $W1$–$W4$ used in Algorithm 3, we use the default value of 1 to evaluate the base design of our BTU-based provisioning approach without any specific emphasis on either the number of instances, scaling operations, queue load or the processing delay.

Besides the configuration aspects for Algorithms 1 and 3, there are also several other configuration aspects for the VISP Runtime. We chose a monitoring timespan of 15 s, i.e., the queue load and resource usage of the system is recorded every 15 s. The resource provisioning interval is set to 60 s. This interval has been selected to update the resource configuration for the SPE in short time intervals while ensuring enough time to download Operator Images from the external repository within one resource provisioning interval.

Regarding the BTU, we use three different BTU durations. The first duration is 60 min (BTU60), which used to be the predominant BTU of Amazon EC2 (https://aws.amazon.com/emr/pricing/). The second duration is 10 min (BTU10), which represented the minimal BTU for the Google Compute Engine (https://cloud.google.com/compute/pricing) and the last duration is 30 min (BTU30), which has been selected to present a middle ground between the other two BTUs. Furthermore, we assume a linear pricing model for the BTUs, i.e., one leasing duration for the BTU10 model results in 1 cost, one leasing duration for the BTU30 model results in 3 cost and the leasing duration for the BTU60 model results in 6 cost. For each data item, which is delayed, we accrue 0.0001 penalty cost, i.e., 10,000 delayed items render the same cost as leasing a VM for 10 min. These penalty cost have been chosen to impose little cost for delayed processing compared to penalty cost in other domains, e.g., for business processes (*Hoenisch et al., 2016*). However, preliminary experiments have shown that higher penalty cost,

e.g., 0.001 or 0.01, would render unreasonable high penalty cost compared to the actual resource cost even for a high SLA compliance. Finally, each Dockerhost VM has the same computational resources with four virtual CPU cores and 7 GB RAM.

## Baseline

To evaluate our BTU-based provisioning approach, we have selected a threshold-based baseline provisioning approach. The baseline implements a commonly used provisioning approach which already achieves very good results in terms of cost reduction against an over-provisioning scenario as shown in our previous work (*Hochreiner et al., 2016a*). The approach considers the amount of data items waiting on the incoming queue for processing as scaling trigger. As soon as the variable $o_{queue}$ exceeds an upper threshold according to Algorithm 1, the SPE triggers an upscaling operation for this operator and as soon as $o_{queue}$ falls below a lower threshold, i.e., 1, the SPE triggers one downscaling action of an operator. Besides the single upscaling trigger, our threshold-based approach triggers the upscaling operation twice, if $o_{queue}$ surpasses a second upper threshold of 250 data items waiting for processing. Regarding the leasing of VMs, we apply an on-demand approach, where the SPE leases a new VM as soon as all currently used VMs are fully utilized and releases a VM, as soon as the last operator instance on that VM is terminated.

## Data arrival pattern

For our evaluation, we have selected four different arrival patterns which simulate different load scenarios for the SPE by submitting varying data volumes to the SPE. The first arrival pattern has three different data volume levels, which are changed stepwise, so that the resulting arrival pattern could be approximated to a sinus curve, as shown in Fig. 7A. These three different volume levels simulate different amounts of manufacturing machines ranging from two to eight machines that emit different amounts of data items, as shown in Table 1. To speed up the evaluation, we simulate the real time data emissions shown in Table 1 every 480 ms. This enables us on the one hand to simulate 500 real-time minutes within only 4 min in the course of our evaluation and therefore also increases the load on the SPE. This also results in a volume level change every 4 min.

The second arrival pattern has only two levels, i.e., the lowest and the highest of the first pattern, which confronts the SPE with more drastic volume changes, as shown in Fig. 7B. Due to the fact that we only apply two different levels, the state changes are twice as long as for the first pattern, i.e., 8 min.

The third and the fourth pattern represent random walks as defined by Eq. (6), whereas $R$ represents a random number between 0 and 1. This random walk is initialized with machine = 4 and we have selected two random walk patterns which stay between one and eight machines. The results of this random walk can be seen in in Fig. 7C for the first random walk and in Fig. 7D for the second one. Due to the random characteristic of the pattern generation, this pattern exhibits more changes of the data volume in short times compared to the first two data arrival patterns.

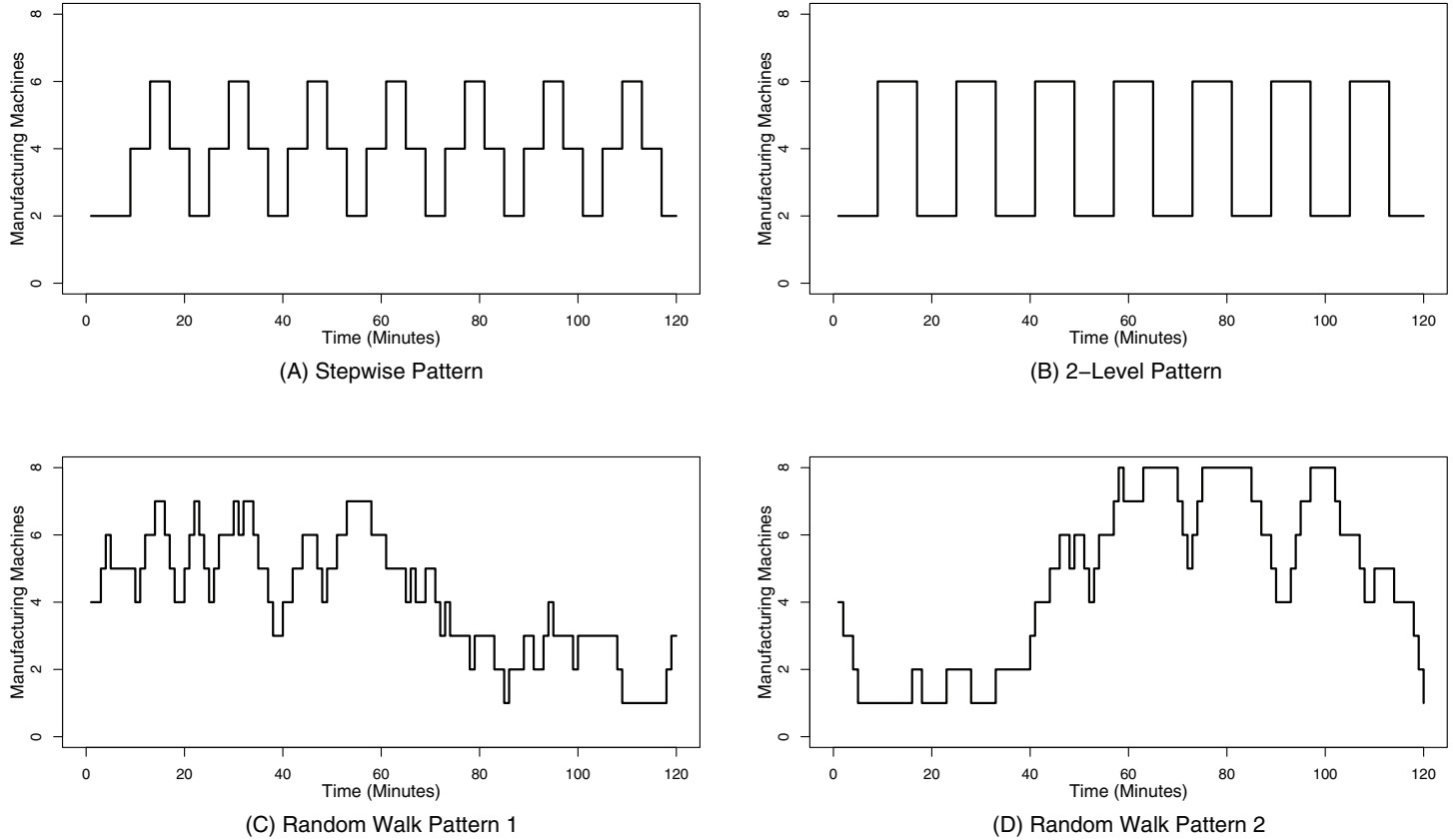

**Figure 7 Data arrival patterns.** (A) shows the stepwise data arrival pattern; (B) shows the 2-level data arrival pattern; (C) shows the data arrival pattern based on the random walk 1; (D) shows the data arrival pattern based on the random walk 2.

$$machine_n = \begin{cases} machine_{n-1} - 1 & R < 0.4 \\ machine_{n-1} & 0.4 \leq R \geq 0.6 \\ machine_{n-1} + 1 & R > 0.6 \end{cases} \tag{6}$$

All four patterns are continuously generated by the VISP Data Provider (https://github.com/visp-streaming/dataProvider) throughout the whole evaluation duration of 120 min.

## Metrics

To compare the evaluation results for both the BTU-based and the threshold-based resource provisioning approaches, we have selected several metrics to describe both the overall cost as well as SLA compliance metrics. After each evaluation run, these metrics are extracted by the VISP Reporting Utility (https://github.com/visp-streaming/reporting). The most important metric is *Paid BTUs*, which describes the total cost for data processing. This value comprises all *VM Upscaling* and *VM Prolonging* operations, which either lease new VMs or extend the leasing for another BTU for existing ones. The *VM*

*Downscaling* sums up all downscaling operations, which are conducted before the end of the BTU.

The next set of metrics describes the SLA compliance of the stream processing application. Each stream processing operator is assigned a specific processing duration which describes the processing duration in a constant over-provisioning scenario. Due to the changing data volume in our evaluation scenarios, it is often the case that the system suffers from under-provisioning for a short time, which results in longer processing durations. To assess the overall compliance of the processing durations, we define three different SLA compliance levels. The first compliance level requires *real-time* processing capabilities, and states the share of data items that are produced within the given processing duration. The second level applies *near-real-time* requirements, which is defined by processing durations that take at most twice as long as the defined processing duration, and the third level applies a *relaxed* strategy, which means that the data items need to be processed within at most five times the stated processing duration. These SLA metrics are obtained from the processing duration of the data items, which are recorded by the operators. To reduce the overall monitoring overhead, we only measure the processing duration of every 10th data item. Nevertheless, preliminary evaluations with other intervals, e.g., every data item or every third data item have shown a similar metric reliability. This similar reliability can be explained due to the fact that observing every 10th data item still yields about 20–40 performance readings/second (depending on the data volume). Therefore it is save to assume that these metrics cover all scaling decisions of the SPE because all other activities, e.g., spawning a new operator instance takes 5–10 s or leasing a new VM takes about 30–60 s.

The *Time to Adapt* metric states the arithmetic mean duration, which is required until the delayed processing for an operator type is back to real-time processing.

The last metrics describe the scaling operations of operator instances. Here we consider *Upscaling*, *Downscaling* as well as *Migration* operations among different hosts.

## RESULTS AND DISCUSSION

For our evaluation we consider four different provisioning approaches. The first approach is the BTU-agnostic threshold-based approach while the other three approaches are BTU-based approaches with three different BTU configurations as discussed in "Evaluation Deployment." To obtain reliable numbers, we have conducted three evaluation runs for each provisioning approach and data arrival pattern, which results in 48 evaluation runs. These evaluations have been executed over the time span of four weeks on a private OpenStack cloud. The raw data of the evaluation runs is hosted on Github (https://github.com/visp-streaming/PeerJ_rawData) and the concrete numbers can be reproduced by means of the VISP Reporting tool (https://github.com/visp-streaming/reporting).

The discussion of our evaluation is divided in four subsections based on the four data arrival patterns. Each subsection features a table which lists the average numbers of the three evaluation runs alongside with their standard deviations. Additionally, we also provide a figure which represents the resource configurations of the operator instances and VMs over the course of the evaluation for each data arrival pattern.

**Table 3 Evaluation results for stepwise scenario.**

| | BTU-based | | | Threshold-based | | |
|---|---|---|---|---|---|---|
| | BTU10 | BTU30 | BTU60 | BTU10 | BTU30 | BTU60 |
| Real-time compliance | 49% ($\sigma$ = 1%) | 52% ($\sigma$ = 1%) | 53% ($\sigma$ = 1%) | 40% ($\sigma$ = 1%) | | |
| Near real-time compliance | 85% ($\sigma$ = 2%) | 90% ($\sigma$ = 1%) | 93% ($\sigma$ = 1%) | 67% ($\sigma$ = 1%) | | |
| Relaxed compliance | 89% ($\sigma$ = 1%) | 93% ($\sigma$ = 1%) | 95% ($\sigma$ = 1%) | 71% ($\sigma$ = 1%) | | |
| Resource cost | 72.33 ($\sigma$ = 3.79) | 92.00 ($\sigma$ = 1.73) | 98.00 ($\sigma$ = 3.84) | 58.00 ($\sigma$ = 1.73) | 79.00 ($\sigma$ = 4.58) | 120.00 ($\sigma$ = 6.00) |
| Real-time total cost | 158.91 ($\sigma$ = 0.82) | 173.39 ($\sigma$ = 0.68) | 174.69 ($\sigma$ = 4.25) | 151.83 ($\sigma$ = 1.95) | 172.83 ($\sigma$ = 4.93) | 213.83 ($\sigma$ = 6.45) |
| Near real-time total cost | 96.85 ($\sigma$ = 0.39) | 108.24 ($\sigma$ = 0.50) | 108.88 ($\sigma$ = 3.17) | 109.59 ($\sigma$ = 1.77) | 130.59 ($\sigma$ = 4.73) | 171.59 ($\sigma$ = 6.35) |
| Relaxed total cost | 90.96 ($\sigma$ = 1.84) | 103.03 ($\sigma$ = 1.04) | 105.41 ($\sigma$ = 2.91) | 102.97 ($\sigma$ = 1.57) | 123.97 ($\sigma$ = 4.49) | 164.97 ($\sigma$ = 6.12) |

For the discussion we are going to analyze the differences between the BTU-based and the threshold-based approach in detail only for the stepwise data arrival pattern because this arrival pattern allows us to isolate specific aspects of the BTU-based approach. Nevertheless, our evaluations shows that the overall trend regarding the SLA compliance and total cost is the same for all four data arrival patterns. For the other arrival patterns we only highlight specific aspects of the individual pattern and refer for all other effects to the discussion of the stepwise data arrival pattern.

**Stepwise data arrival pattern**

For the stepwise pattern, we can see that the overall SLA compliance is higher for the BTU-based approach for all three SLA compliance scenarios as shown in Table 3. This compliance benefit ranges from 9% for the BTU10 configuration in the real-time compliance scenario, up to 24% in the relaxed compliance scenario for the BTU60 configuration. The SLA compliance gain can be explained due to the downscaling strategy of the BTU-based approach in contrast to the on-demand one for the threshold-based approach. The threshold-based approach only considers the amount of data items that are considered for processing based on each operator type for the scaling decisions, which can be observed in Fig. 8D. This figure shows that the line for the operator instances follows the data volume very closely with a short delay because the threshold-based approach can only react based on the changes of the data volume. On closer inspection, one can also identify smaller increases after the downscaling phase, e.g., around minutes 40, 55 or 70. These smaller bumps indicate that the downscaling approach was too eager and the SPE has to compensate it by scaling up again. Throughout this time span, i.e., between the detection of a lack of processing capabilities and the successful upscaling for the operator type, the SPE is very likely to violate the SLA compliance, especially in the real-time scenario.

The BTU-based approach does not exhibit such a strongly coupled relationship between the operator instances and the data volume. While the upscaling trigger is the same for both scenarios, there are clear differences in the downscaling behavior. The BTU-based approach only considers downscaling activities briefly before the end of a BTU, e.g., around minutes 20 or 40 for the BTU10 scenario, around minute 30 for the BTU30 scenario and around minute 60 for the BTU60 scenario in Figs. 8B and 8C. The result of

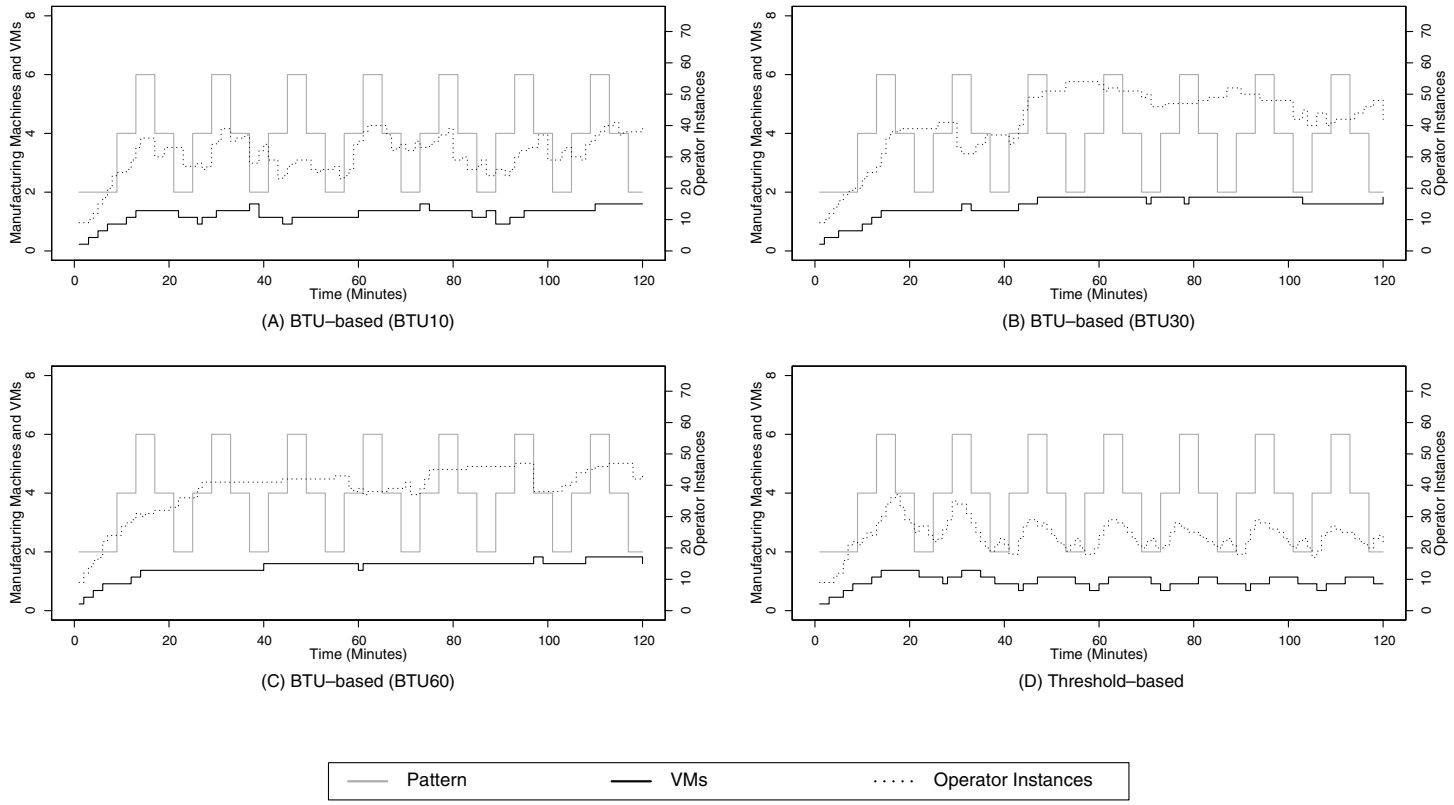

**Figure 8 Stepwise pattern.** (A) shows the resource provisioning configuration using the BTU-based approach using a BTU of 10 min for the stepwise data arrival pattern; (B) shows the resource provisioning configuration using the BTU-based approach using a BTU of 30 min for the stepwise data arrival pattern; (C) shows the resource provisioning configuration using the BTU-based approach using a BTU of 60 min for the stepwise data arrival pattern; (D) shows the resource provisioning configuration using the threshold-based approach for the 2-level data arrival pattern.

this lazy downscaling strategy is a decrease of scaling activities, especially for the BTU30 and BTU60 scenario. This decrease in scaling activities results in a better SLA compliance since the SPE already maintains the processing capabilities for future data volume peaks as this is the case for the stepwise data arrival pattern. This results in high SLA compliance values of over 90% for the BTU30 and BTU60 scenario. It needs to be noted that the lack of active downscaling activities does not increase the cost for computational resources since these resources have already been paid at the beginning of their BTU.

The BTU-based downscaling operations are often triggered at suitable times, e.g., around minutes 20 and 38 for the BTU10 configuration or minute 70 for the BTU30 configuration, where the downscaling activities do not impact the SLA compliance. Nevertheless, there are also points in time, when the BTU of a VM coincides with a peak of the data volume, e.g., at minute 30 for the BTU30 configuration. In these situations, the BTU-based approach will initialize the downscaling procedure to release a VM shortly before the end of its BTU. In this specific case around minute 30 for the BTU30 scenario, the downscaling procedure is successful because monitoring does not report any delays for processing based on Algorithm 3 and the VM is triggered to be shut

down. But in the next reasoning cycle, the SPE realizes the lack of processing capabilities and leases another VM to compensate the resource requirements. Although these non-efficient scaling operations result in a measurable overhead as well as an SLA compliance reduction, the BTU-based approach still achieves a better SLA compliance than the threshold-based approach.

Furthermore, it can be seen that the amount of scaling activities for the operator instances is inverse to the length of the BTU. For the BTU10 configuration, it can be observed in Fig. 8A that the level of scaling activities is similar to those of the threshold-based scenario. This results in a rather low SLA compliance, but for the BTU30 and especially the BTU60 there are less downscaling events, i.e., BTU ends, which reduces the need to scale up again to comply with future data volume peaks.

Besides the SLA compliance, we also consider the operational cost for data processing. These cost are composed of the resource cost, i.e., the cost for leasing VMs and the penalty cost, which accrue for delayed data processing. In Table 3, it can be seen that the resource cost for the BTU10 and BTU30 configuration are higher than the ones for the threshold-based ones. These higher cost can be explained due to the defensive approach of releasing VMs for the BTU-based approach, which often results in leasing the VM for another BTU based on Algorithm 3. For the BTU60 configuration, the resource cost are around 19% lower than those for the threshold-based configuration. Although the BTU60 configuration uses more computational resources, as shown in Fig. 8C, the overall cost are lower, because the threshold-based approach releases the VMs often prematurely before the end of their BTU, which results in a waste of already paid resources.

When we consider only the resource cost, we can see that the BTU-based approach only outperforms the threshold-based approach for the BTU60 configuration. Nevertheless, this situation changes when we also consider the penalty cost, i.e., 1 cost for 10,000 delayed items. After adding the penalty cost and analyzing the total cost for the different compliance scenarios, we can see that only the real-time total cost for the BTU10 configuration is higher than the threshold-based approach. All other scenarios result in slightly less cost for the BTU30 configuration in the real-time scenario and up to a 36% cost-reduction for the near real-time one for the BTU60 configuration.

## Two-level data arrival pattern

The two-level data arrival pattern exhibits the same trend for the SLA compliance and cost for the stepwise data arrival pattern as shown in Table 4. When we analyze the Figs. 9A–9D, we can also see a similar scaling behavior compared to the stepwise data arrival pattern. Nevertheless, there is one notable effect for the BTU60 configuration in Fig. 9C. The BTU-based provisioning approach tends to start more and more operator instances throughout the evaluation run. We can see that after minute 20, when the SPE has enough processing capabilities, the upscaling trigger requests new operator instances from time to time to cope with the data volume. These upscaling operations are most likely due to minor external events, e.g., short network delays due to other applications running on the same physical hardware, which causes the SPE to obtain

**Table 4 Evaluation results for two-level scenario.**

| | BTU-based | | | Threshold-based | | |
|---|---|---|---|---|---|---|
| | BTU10 | BTU30 | BTU60 | BTU10 | BTU30 | BTU60 |
| Real-time compliance | 48% (σ = 1%) | 50% (σ = 1%) | 55% (σ = 1%) | 40% (σ = 2%) | | |
| Near real-time compliance | 84% (σ = 2%) | 88% (σ = 0%) | 93% (σ = 2%) | 68% (σ = 2%) | | |
| Relaxed compliance | 88% (σ = 2%) | 91% (σ = 0%) | 95% (σ = 1%) | 72% (σ = 2%) | | |
| Resource cost | 82.33 (σ = 5.13) | 96.00 (σ = 7.94) | 104.00 (σ = 6.93) | 66.00 (σ = 0.00) | 86.00 (σ = 1.73) | 122.00 (σ = 3.46) |
| Real-time total cost | 169.17 (σ = 6.83) | 177.62 (σ = 6.82) | 175.90 (σ = 4.77) | 157.88 (σ = 2.64) | 177.88 (σ = 4.17) | 213.88 (σ = 4.16) |
| Near real-time total cost | 108.35 (σ = 6.74) | 155.43 (σ = 8.18) | 114.50 (σ = 6.28) | 114.62 (σ = 3.21) | 134.62 (σ = 4.19) | 170.62 (σ = 2.94) |
| Relaxed total cost | 102.37 (σ = 6.74) | 110.62 (σ = 8.18) | 111.73 (σ = 6.28) | 108.83 (σ = 2.40) | 128.83 (σ = 3.37) | 164.83 (σ = 2.59) |

**Figure 9 Two-level pattern.** (A) shows the resource provisioning configuration using the BTU-based approach using a BTU of 10 min for the 2-level data arrival pattern; (B) shows the resource provisioning configuration using the BTU-based approach using a BTU of 30 min for the 2-level data arrival pattern; (C) shows the resource provisioning configuration using the BTU-based approach using a BTU of 60 min for the 2-level data arrival pattern; (D) shows the resource provisioning configuration using the threshold-based approach for the 2-level data arrival pattern.

new processing capabilities. The result of this slow increase of operator instances over time is that the SPE is likely to have more processing capabilities than it actually needs. Nevertheless, at the end of the BTU of a VM, the necessity of these processing capabilities is evaluated, and for example in the BTU60 configuration, the operator instances are cut back around minute 60. After a short recalibration phase between

**Table 5 Evaluation results for random walk 1.**

| | BTU-based | | | Threshold-based | | |
|---|---|---|---|---|---|---|
| | BTU10 | BTU30 | BTU60 | BTU10 | BTU30 | BTU60 |
| Real-time compliance | 49% (σ = 1%) | 52% (σ = 2%) | 54% (σ = 0%) | 39% (σ = 0%) | | |
| Near real-time compliance | 85% (σ = 2%) | 90% (σ = 3%) | 93% (σ = 0%) | 66% (σ = 1%) | | |
| Relaxed compliance | 89% (σ = 2%) | 93% (σ = 2%) | 95% (σ = 1%) | 71% (σ = 1%) | | |
| Resource cost | 69.33 (σ = 3.51) | 95.00 (σ = 1.73) | 110.00 (σ = 3.46) | 61.33 (σ = 1.53) | 86.00 (σ = 1.73) | 128.00 (σ = 6.93) |
| Real-time total cost | 158.19 (σ = 4.99) | 176.94 (σ = 4.70) | 185.95 (σ = 3.40) | 158.68 (σ = 1.54) | 183.35 (σ = 1.40) | 225.35 (σ = 6.56) |
| Near real-time total cost | 94.44 (σ = 6.14) | 111.43 (σ = 5.65) | 121.61 (σ = 2.89) | 115.55 (σ = 1.14) | 140.22 (σ = 1.77) | 182.22 (σ = 6.88) |
| Relaxed total cost | 88.11 (σ = 5.86) | 106.67 (σ = 4.69) | 117.91 (σ = 3.16) | 107.54 (σ = 1.47) | 132.21 (σ = 2.14) | 174.21 (σ = 7.25) |

minutes 65 and 75, the SPE follows the same pattern again until the resources are cut back again around minute 120. This mechanism allows the SPE to use the already leased resources, i.e., no additional VMs are leased from minute 80 until 120, to achieve a high resource utilization.

### Random walk 1 data arrival pattern

Based on the numbers of Table 5, we can see that the random walk 1 data arrival pattern follows the same trend for the SLA compliance as well as total cost as the stepwise data arrival pattern. At closer inspection we can see that the SLA compliance is very similar with a deviation of less than 3%. This aspect shows that both the baseline as well as the BTU-based provisioning approach have similar characteristics for the rather simple data arrival pattern, like the stepwise or two-level one, as well as random ones.

Based on the Figs. 10A–10D, we can identify one notable difference between the BTU-based and the threshold-based resource provisioning approach. While the operator instance curve and the data volume curve are well aligned for the threshold-based and the BTU10 configuration, we can identify a clear gap for the BTU30 in Fig. 10B and especially for the BTU60 configuration (Fig. 10C). For the latter two configurations, the operator instance curve remains high although the data volume decreases over time. This behavior can be explained due to the optimal resource usage of the already paid resources, which enables the BTU30 and BTU60 configuration to keep the running operator instances without any additional cost. Although this behavior may seem to be a waste of resources at first sight due to the deviation of the actual data volume and the operator instances, it becomes beneficial for the SPE in terms of SLA compliance when the volume rises again, e.g., around minutes 85 or 120.

### Random walk 2 data arrival pattern

The numerical results in terms of the SLA compliance and total cost follow similar trends as for the stepwise data arrival pattern scenario, based on the numbers in Table 6. For this data arrival pattern also only the BTU10 configuration requires more cost than the threshold-based baseline for the real-time scenario. All other configurations and scenarios result in lower cost than the baseline. When we analyze the graphical representation of Figs. 11A–11D

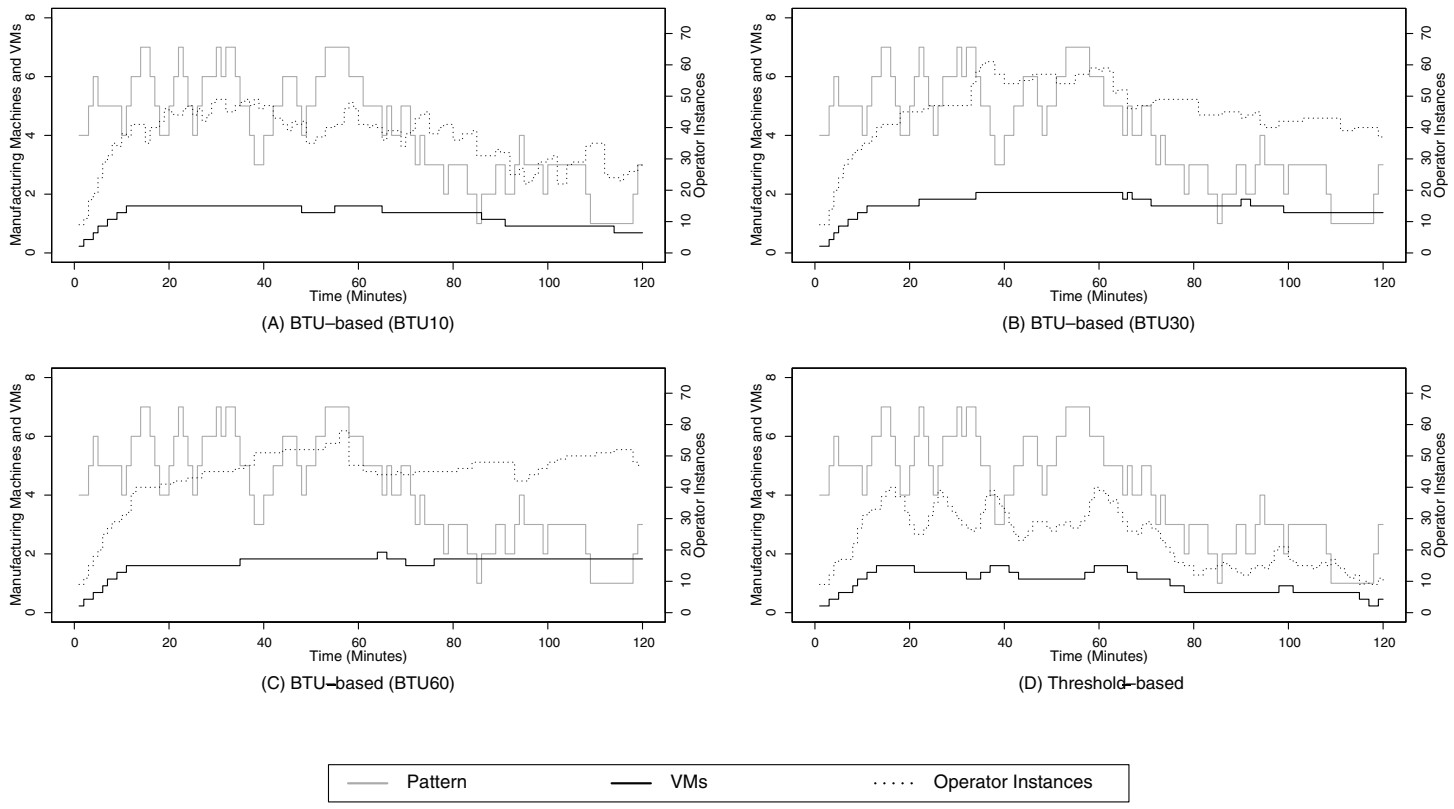

**Figure 10 Random walk pattern 1.** (A) shows the resource provisioning configuration using the BTU-based approach using a BTU of 10 min for the random walk data arrival pattern 1; (B) shows the resource provisioning configuration using the BTU-based approach using a BTU of 30 min for the random walk data arrival pattern 1; (C) shows the resource provisioning configuration using the BTU-based approach using a BTU of 60 min for the random walk data arrival pattern 1; (D) shows the resource provisioning configuration using the threshold-based approach for the random walk data arrival pattern 1.

**Table 6 Evaluation results for random walk 2.**

|  | BTU-based | | | Threshold-based | | |
|---|---|---|---|---|---|---|
|  | BTU10 | BTU30 | BTU60 | BTU10 | BTU30 | BTU60 |
| Real-time compliance | 49% ($\sigma$ = 2%) | 51% ($\sigma$ = 1%) | 53% ($\sigma$ = 0%) | 41% ($\sigma$ = 1%) | | |
| Near real-time compliance | 87% ($\sigma$ = 2%) | 90% ($\sigma$ = 1%) | 92% ($\sigma$ = 1%) | 70% ($\sigma$ = 1%) | | |
| Relaxed compliance | 90% ($\sigma$ = 2%) | 94% ($\sigma$ = 1%) | 95% ($\sigma$ = 0%) | 75% ($\sigma$ = 75%) | | |
| Resource cost | 74.00 ($\sigma$ = 6.08) | 90.00 ($\sigma$ = 0.00) | 106.00 ($\sigma$ = 3.46) | 59.67 ($\sigma$ = 4.04) | 82.00 ($\sigma$ = 9.17) | 118.00 ($\sigma$ = 9.17) |
| Real-time total cost | 172.67 ($\sigma$ = 5.35) | 184.03 ($\sigma$ = 0.99) | 195.73 ($\sigma$ = 3.33) | 164.18 ($\sigma$ = 3.84) | 187.41 ($\sigma$ = 9.88) | 223.41 ($\sigma$ = 9.88) |
| Near real-time total cost | 100.17 ($\sigma$ = 6.65) | 108.91 ($\sigma$ = 2.47) | 120.59 ($\sigma$ = 3.59) | 113.41 ($\sigma$ = 3.47) | 135.98 ($\sigma$ = 9.93) | 171.98 ($\sigma$ = 9.93) |
| Relaxed total cost | 92.67 ($\sigma$ = 7.15) | 101.96 ($\sigma$ = 1.61) | 115.67 ($\sigma$ = 3.48) | 104.43 ($\sigma$ = 2.49) | 127.19 ($\sigma$ = 8.29) | 163.19 ($\sigma$ = 8.29) |

for the random walk 2 data arrival pattern, the most prominent difference in contrast to the random walk 1 data arrival pattern is the even better alignment of the operator instance and data volume curves. This is due to the fact that the data volume is rising for the second part of the evaluation, i.e., after minute 40, and the already paid resources can be actively used for data processing instead of only serving as free backup processing capabilities.

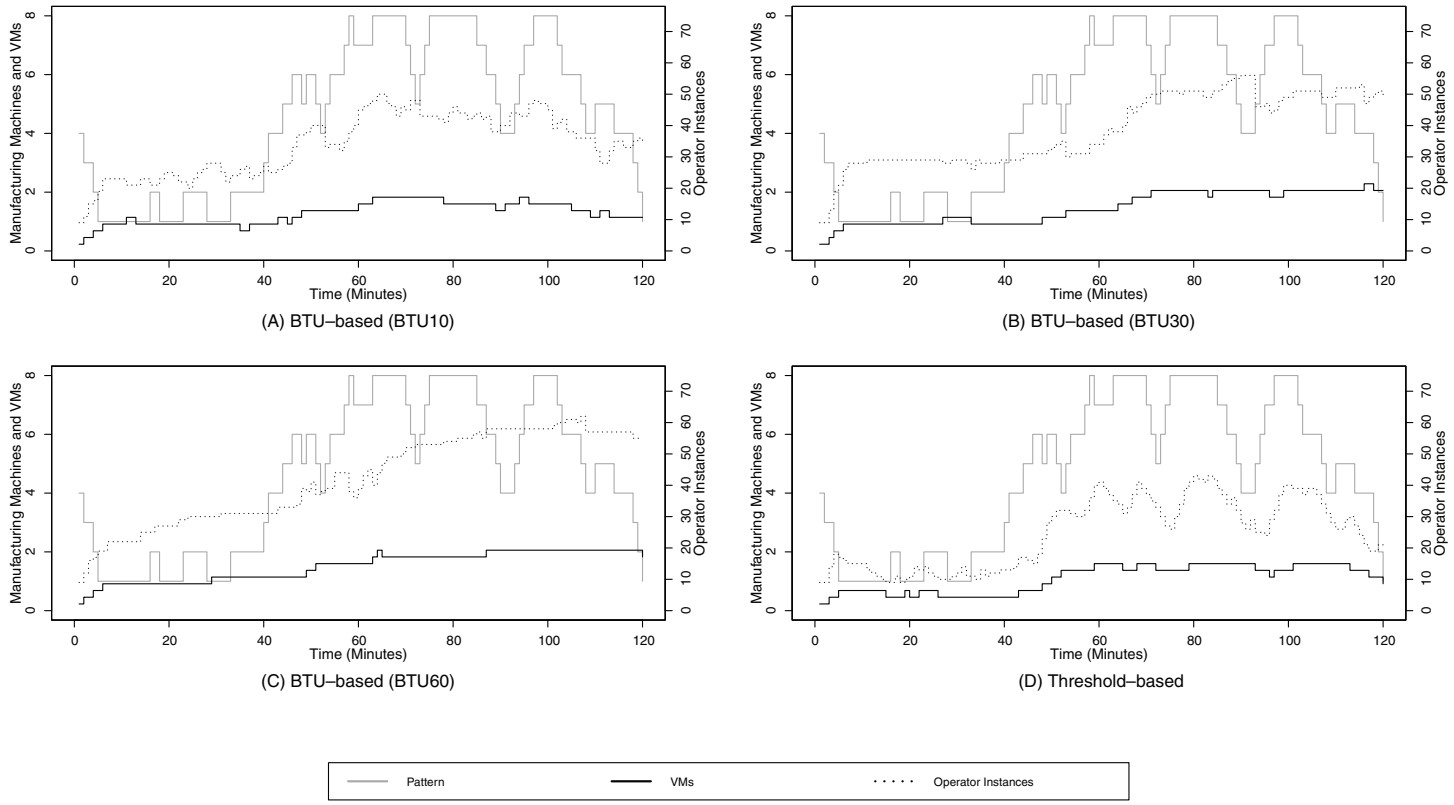

**Figure 11 Random walk pattern 2.** (A) shows the resource provisioning configuration using the BTU-based approach using a BTU of 10 min for the random walk data arrival pattern 2; (B) shows the resource provisioning configuration using the BTU-based approach using a BTU of 30 min for the random walk data arrival pattern 2; (C) shows the resource provisioning configuration using the BTU-based approach using a BTU of 60 min for the random walk data arrival pattern 2; (D) shows the resource provisioning configuration using the threshold-based approach for the random walk data arrival pattern 2.

Furthermore, it can be seen that the BTU-based approach requires less scaling activities between minute 60 and 120 in contrast to the threshold-based approach in Fig. 11D. This is again due to the lazy release characteristics of the BTU-based approach, which result in a higher SLA compliance in contrast to the threshold-based approach.

## Evaluation conclusion

When we compare the evaluation results of the four different data arrival patterns, we can see that they all share the same trend. Regarding the SLA compliance, we can see that the BTU-based approach achieves a better SLA compliance for all configurations for all compliance scenarios. Furthermore, the SLA values are roughly the same (with a maximum deviation of 3%) across all data arrival patterns despite their different characteristics.

For the total cost, we can also see that only the BTU10 configuration for the real-time scenario results in higher cost in contrast to the baseline. All other configurations and scenarios for the BTU-based approach exhibit a cost reduction. Additionally it must be noted that the resource cost are always lower for the BTU60 configuration than for the threshold-based approach.

We can also observe that the compliance for real-time data processing on cloud infrastructures is rather low, i.e., around 40% for the baseline and around 50–55% for the BTU-based approach. This is mainly due to the fact that cloud environments are often influenced by other applications running on the same physical hardware. This can result in minor data transmission or processing delays that have a severe impact on the SLA compliance. Nevertheless, we can see that for the near real-time and relaxed time scenarios, the SLA compliance ranges from 84% to 95% for the BTU-based approach, which meets the requirements of our motivational scenario discussed in "Motivation."

## Threats to applicability

Although the presented system model builds on top of real-world observations, it cannot be guaranteed that all external aspects are adequately considered in our system model which may result in a non-optimal performance in real-world deployments. Nevertheless, we consider this risk as rather small, since we have already conducted our evaluations in a cloud-based testbed, which already considers external influences by other applications running on the same cloud environment. To consider such external effects for the evaluation, we repeated each evaluation scenario and configuration three times on different days (including the weekend) to cover different usage scenario on the OpenStack-based cloud due to other stakeholders on the same physical hardware.

## RELATED WORK

In the last couple of years, the landscape of SPEs has been constantly increasing. In contrast to the rather basic SPEs, like Aurora (*Balakrishnan et al., 2004*) or Borealis (*Abadi et al., 2005*), which have been designed more than a decade ago, today's SPEs incorporate technological advances like cloud computing and can process large volumes of data in parallel. While some of these SPEs are rather focused on cluster-based deployments, like System S (*Gedik et al., 2008*), most are designed to utilize cloud-based deployments, like Apache Spark (*Zaharia et al., 2010*), Apache Flink (*Carbone et al., 2015*), Apache Storm (*Toshniwal et al., 2014*) or its derivative Heron (*Kulkarni et al., 2015*). Despite the focus on designing efficient SPEs, the resource elasticity aspects of individual operator instances (or workers) have only been picked up recently, e.g., for Apache Spark Streaming or the automatic reconfiguration for Apache Storm based on hints for the number of workers. To the best of our knowledge there exists no established SPEs which consider a two-level resource provisioning architecture, since most SPEs outsource this functionality to other frameworks like Apache Mesos or Kubernetes. However, there are a couple of prototypes and concepts in the literature, which propose a mechanism for elastic stream processing.

Several research groups have picked up the challenge of replacing the previously dominant strategy of data quality degradation, i.e., load shedding (*Babcock, Datar & Motwani, 2004*; *Tatbul, Çetintemel & Zdonik, 2007*), with resource elasticity. Nevertheless, most of the first publications focus on an optimal resource configuration only when deploying a topology and do not consider any updates at runtime, e.g., *Setty et al. (2014)* for pub/sub systems or *Florescu & Kossmann (2009)* for database systems. The next step

toward resource elasticity was proposed by *Lim, Han & Babu (2013)*, who proposed the redeployment of complete applications for database management systems whenever the resource requirements change. Although this approach already supports resource elasticity, it was required to refine this approach to only consider operator instances instead of complete applications. One of the first publications in the domain of data stream processing was authored by *Schneider et al. (2009)*, which proposed the parallelization of stream processing operations with System S. Because this first approach only considered stateless operators, the authors complemented their approach in a succeeding publication to consider the replication of stateful operators (*Gedik et al., 2014*). Besides the elasticity extension to System S, there are also several proposed extensions to Apache Storm, which replace the default scheduler with custom implementations to optimize the parallelization of operators as well as the placement thereof on different computational resources. Two of these approaches have been presented by *Aniello, Baldoni & Querzoni (2013)* and *Xu et al. (2014)*. These two publications present threshold-based custom schedulers, which can adopt the topology deployment at runtime, depending on the incoming data volume and the actual load for Apache Storm. Although any replication of a specific operator provides additional processing capabilities, it needs to be noted that any reconfiguration of the topology enactment has a negative impact on the processing performance. To minimize these reconfiguration aspects, Stela (*Xu, Peng & Gupta, 2016*), introduces new performance indicators to focus on the actual throughput of the SPE and to reduce any reconfiguration aspects.

To extend the rather static aspect of the threshold-based scaling approaches, *Heinze et al. (2015)* propose a threshold-based resource optimization, whose thresholds are adopted based on an online learning mechanism within a custom SPE. This allows resource optimization to adapt the otherwise fixed thresholds, which are predefined before the topology enactment, at runtime to improve the resource utilization based on actual monitoring data. SEEP (*Castro Fernandez et al., 2013*), another custom SPE, also proposes a simple threshold-based replication mechanism. In contrast to the other already discussed approaches, SEEP focuses on stateful operators and employs a dedicated fault tolerance mechanism.

Besides the basic replication approaches, there are also some works that optimize specific aspects for the topology enactment. One of these aspects is the partitioning of data to optimize the data flow among the operators, especially regarding stateful operators. The Streamcloud (*Gulisano et al., 2012*) SPE proposes a mechanism to partition the incoming data to distribute it efficiently among different replicas of one operator type. Another approach for optimizing the overall efficiency of a topology enactment is to optimize the placement of operators within a potential heterogeneous pool of computational resources. *Cardellini et al. (2015)* propose an extension to Apache Storm, which considers an optimal placement of operators in terms of SLA-based criteria on different cloud resources. Furthermore, *De Matteis & Mencagli (2016)* present a predictive approach to minimize the latency and improve the energy efficiency of the SPE. This approach allows to reduce the reconfiguration of SPEs, which is also one of the objectives in our approach. The last notable approach for optimizing the topology enactment on

cloud resources is to optimize the deployment of operators according to their specific processing tasks. *Hanna et al. (2016)* consider different types of VMs, e.g., with an emphasis on CPU or GPU, and optimize the deployment based on the suitability of these machines to conduct specific operations, e.g., matrix multiplications are significantly faster when executed on the GPU.

Although the literature already provides different optimization approaches, to the best of our knowledge, none of these approaches considers the BTU aspect of VMs when optimizing processing resources as proposed in this paper. Also, most of the discussed approaches only aim at optimizing the amount of replicas for processing operators, but do ignore the reconfiguration overhead during the topology enactment.

## CONCLUSION

Within this paper, we have discussed the most important requirements for optimizing data stream processing in volatile environments. Based on these requirements, we have developed an extensive system model for which we have presented a BTU-based optimization approach. This optimization approach has been evaluated with four different data arrival pattern against a threshold-based approach, which already provides a significant cost reduction based on our previous work (*Hochreiner et al., 2016a*). The evaluation has shown that the BTU-based approach results in a better SLA compliance which also achieves a better overall cost structure compared to the threshold-based approach.

Nevertheless, as a result of the evaluation, we have also identified a potential extension possibility for our BTU-based approach, namely the addition of a more sophisticated predictive component. So far we only consider the trend for upscaling operator instances, but we do not consider historical information nor other monitoring information, e.g., as suggested by *Copil et al. (2016)*, for downscaling purposes, which could yield even better results. In our future work, we plan to also apply our BTU-based approach to hybrid clouds. This requires an extension of the optimization model regarding the networking capabilities among these clouds. Furthermore, we plan to investigate the structural properties of the topology in more detail, e.g., to identify critical paths or high volume operators, such as the operators O2 and O6 in our topology. These insights may help us to apply different scaling priorities, especially for downscaling operations to avoid oscillating effects. In addition we also plan to evaluate the impact of using the individual weights $W1-W4$ in Algorithm 3 within both private and hybrid clouds.

### Funding

This work is supported by TU Wien research funds and by the Commission of the European Union within the CREMA H2020-RIA project (Grant agreement no. 637066). There was no additional external funding received for this study. The funders had no role in study design, data collection and analysis, decision to publish, or preparation of the manuscript.

## Grant Disclosures

The following grant information was disclosed by the authors:

TU Wien research funds and by the Commission of the European Union within the CREMA H2020-RIA project: 637066.

## Competing Interests

Schahram Dustdar and Stefan Schulte are Academic Editors for PeerJ.

## Author Contributions

- Christoph Hochreiner conceived and designed the experiments, performed the experiments, analyzed the data, contributed reagents/materials/analysis tools, wrote the paper, prepared figures and/or tables, performed the computation work, reviewed drafts of the paper.
- Michael Vögler conceived and designed the experiments, wrote the paper, reviewed drafts of the paper.
- Stefan Schulte wrote the paper, reviewed drafts of the paper.
- Schahram Dustdar reviewed drafts of the paper.

## Data Availability

The source code repositories for the prototype implementation, evaluation tools as well as the raw data for the evaluation can be found in the VISP-streaming project: https://github.com/visp-streaming.

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
