# Peer review of "Cost-efficient enactment of stream processing topologies"

_PeerJ Computer Science, doi:10.7717/peerj-cs.141_

## Round 0.1 · original submission · Major Revisions

in light of the reviewer comments it seems a good idea to double check a couple of things specifically the drift that occurred between title, abstract and paper content

Reviewer 1 ·

Basic reporting

The authors present an stream processing engine that uses docker containers on top of VM's. Docker containers promise to provide more flexible deployment of stream operators under varying data arrival scenarios.

A deployment algorithm has to optimize resource usage while taking SLA requirements into account. The authors present a heuristic to this NP-hard optimization problem. The authors evaluate their novel BTU-based optimization approach with a more naive threshold-based approach. They can show that their new method is more cost efficient.

Experimental design

For the evaluation the authors simulate input data (i.e. sensor data) using tools from their VISP Testbed. Their stream processing engine is runs on a private OpenStack deployment. For the evaluation the authors compare their topology enactment algorithm that optimizes Billing Time Units with a simple algorithm that uses thresholds on the incoming queues to determine if VM's should be added or removed to the tolopogy. The experiments are performed with different arrival patterns and variable data volume levels. For comparison the the authors compute various metrics, including paid Billing Time Units and various Quality-of-Service metrics. The author convincingly performed the evaluation on a near real world example (real-world example but simulated data) using variable data load scenarios. The evaluation metrics are reasonable and provide insights into the behavior of their algorithm.

Validity of the findings

no comment

Additional comments

This is a well written paper!

I miss some discussion regarding I/O. How would a high IO to compute ratio impact the deployment strategy?

It is not clear to me if the software the authors developed is available for use, ideally as open source.

Reviewer 2 ·

Basic reporting

* the paper is clearly structured and easy to read
* sufficient background is presented and the studied related work meets the expected standards
* most figures are easy to read, except Figure 8, which is impossible to read on b/w printout and Figure 9 and 10, which are totally overloaded

Experimental design

* the paper studies a very interesting topic (elastic scheduling considering the monetary cost), which is novel and not studied in detail in the existing related work

* the methods are clearly described and easy to follow

Negative points:
* I see a clear gap between the title of the paper, its abstract, and content:
** the abstract and the paper itself are focusing on different goals. While the abstract speaks about "maximizing the resource usage", the paper tries to minimize the costs, although the authors point out that these goals are not necessarily exchangable
** the title contains "utilizing container technologies", I don't see how a container technology helps to solve the problem described and what is the difference to the use of a thread / process per operator like previous solutions do;
for me the container technology is an implementation detail and not relevant for the contribution of the paper

* I don't understand the described approach completely and have multiple questions:
** how do the algorithms Algorithm 1,2, and 3 contribute to the goal of minimizing the number of reconfigurations and the overall cost? I understand that these algorithms choose the optimal parallelization degree per operator,
but not how they influence the cost or minimize the number of reconfigurations. The authors should describe this more clearly.
** the core contribution for me is the approach illustrated in Figure 5, but this figure is only described very shortly and informal. Moreover, the approach provided in this Figure answers the problem only partially:
E.g. shouldn't you undo all changes if you can not release the host in order to minimize the number of reconfigurations?

* All described algorithms have "magic" values including "scalingThreshold" (Algorithm 1), CF (Algorithm 2), QL, W1, W2, W3, and W4, which are not discussed or evaluated in detail.

Based on the points raised above, I cannot judge if the approach above really solves the described challenge or if the measured cost reduction is only a positive "side effect".

Validity of the findings

* The used workload pattern are too simplistic for a meaningful evaluation and the authors themself raise some concerns regarding practical applicability (line 658ff).

Additional comments

Minor points:
* line 67/68: I would recommend to add Apache YARN and Kubernetes to this list: YARN is the most important cluster manager used in practice today and Kubernetes gets more and more importance overall
* line 69ff: I totally disagree to this statement. Mesos and similar frameworks do not decide how much resources are used by the data streaming system. Both rely on a two-level architecture,
where the framework like Spark, Storm, or Flink decides how much resources is needed. YARN or Mesos only ensures all running frameworks get sufficient resources and fairly share them between multiple users.
For this task, simple methods like BinPacking work quite well (especially given the intended scalability of Mesos or YARN).
* line 661ff: Also in private cloud environments cost efficiency is important to the user. Typically, the resource costs are cross-charged to different business units of the same company and you need to minimize the cost for your department.
* line 670ff: I disagree that elasticity has not reached established products: Spark Streaming has Dynamic Allocation (https://issues.apache.org/jira/browse/SPARK-12133),
Flink supports elastic scaling of stateless flows, Heron recently published a paper on elastic scalability on VLDB2017, and also some of the research about System S reached the final product.
* your related work does not reflect any works on cost efficiency of data management solutions, you can consider referencing one of the following:
** Florescu et al. "Rethinking Cost and Performance of Database Systems.", SIGMOD Records 2009.
** Kossmann et al. "An Evaluation of Alternative Architectures for Transaction Processing in the Cloud.", SIGMOD 2010.
** Lim et al. "How to fit when no one size fits.", CIDR 2013.
** Kllapi et al. "Schedule Optimization for Data Processing Flows on the Cloud.", SIGMOD 2011.
** Setty et al. "Cost-Effective Resource Allocation for Deploying Pub/Sub on Cloud.", ICDCS 2014.

---

## Round 0.2 · accepted · Accept

I agree with the reviewer (who had previously requested a major revision of your work) that it is now Acceptable. Please consider the additional suggestions they made.

Reviewer 2 ·

Basic reporting

* the paper is clearly structured and easy to read
* sufficient background is presented and the studied related work meets the expected standards
* all figures and tables are easy to read and understand
* the presented level of detail about related work meets the expected quality

Experimental design

* the paper studies a very interesting topic (elastic scheduling considering the monetary cost), which is novel and not studied in detail in the existing related work
* the methods are clearly described and easy to follow
* all my previous comments regarding the description of the approach, its parameters and the title of the work have been addressed

Validity of the findings

* the evaluation scenarios now meet the state of the art and allow to judge the quality of the proposed solution

Additional comments

I think the paper has improved significantly since the earlier version I reviewed and can now be published. A few things the authors could consider for the final version of the paper:

* I personally would prefer to use another kind of measure for the QoS, which is more familiar to the generic reader. You could replace your compliance measure by different latency percentiles e.g. the 99th, 98th and 90th percentile of the end to end latency
* you motivate the used BTU interval 60 minutes with the fact that AWS uses the same interval. Unfourtnately, AWS changed its price model during your paper was in review [1]. You maybe should consider to remove this statement from your paper.

[1] https://aws.amazon.com/blogs/aws/new-per-second-billing-for-ec2-instances-and-ebs-volumes/